# Dual-Branch Representations with Dynamic Gated Fusion and Triple-Granularity Alignment for Deep Multi-View Clustering

**Wenyuan Kong**
Independent Researcher
Beijing, China
kongwenyuan97@gmail.com

**Zhibin Gu**[*]
College of Computer and Cyber Security
Hebei Normal University, Shijiazhuang, China
guzhibin@hebtu.edu.cn

**Bing Li**
School of Computer Science, China University of Labor Relations, Beijing, China
State Key Laboratory of Multimodal Artificial Intelligence Systems, Institute of Automation
Chinese Academy of Sciences, Beijing, China
bli@nlpr.ia.ac.cn

## Abstract

Multi-view clustering seeks to exploit complementary information across different views to enhance clustering performance, where both semantic and structural information are crucial. However, existing approaches often bias toward one type of information while treating the other as auxiliary, overlooking that the reliability of these signals may vary across datasets and that semantic and structural cues can provide complementary and parallel guidance. As a result, such methods may face limitations in generalization and suboptimal clustering performance. To address these issues, we propose a novel method, **D**ual-branch **R**epresentations with dynamic gat**E**d fusion and triple-gr**A**nularity align**M**ent (**DREAM**), for deep multi-view clustering. Specifically, DREAM disentangles semantic information via a Variational Autoencoder (VAE) branch, while simultaneously captures structure-aware features through a Graph Convolutional Network (GCN) branch. The resulting representations are dynamically integrated using a gated fusion module that leverages structural cues as complementary guidance, adaptively balancing semantic and structural contributions to produce clustering-oriented latent embeddings. To further improve robustness and discriminability, we introduce a triple-granularity feature alignment mechanism that enforces consistency across views, within individual samples, and intra-cluster, thereby preserving semantic-structural coherence while enhancing inter-cluster separability. Extensive experiments on benchmark datasets demonstrate that DREAM significantly outperforms SOTA approaches, highlighting the effectiveness of disentangled dual-branch encoding, adaptive gated fusion, and triple-granularity feature alignment.

## 1 Introduction

Recent advances in sensing and Internet technologies have enabled the collection of data from multiple sources, offering diverse and complementary information about the same phenomenon (Fang et al., 2023; Zhou et al., 2024). For example, rasterized high-definition map and LiDAR data provide distinct yet complementary views for autonomous driving (Fadadu et al., 2022). Multi-view clustering (MVC), which aims to exploit both the shared and complementary information across views to uncover the underlying pattern of samples, has therefore emerged as a crucial paradigm for analyzing complex multi-modal data. In recent years, MVC has achieved remarkable success across domains, such as computer vision (Wang et al., 2024), biomedicine (Rappoport & Shamir, 2018)

---

[*]Corresponding author.

and social interactions (Yang et al., 2014), where clustering performance has been substantially improved by integrating heterogeneous perspectives.

Depending on the underlying learning paradigms, MVC methods can be broadly categorized into conventional (shallow) and deep learning-based approaches. Conventional methods, including non-negative matrix factorization (NMF) (Liu et al., 2013), multi-kernel clustering (Zhang et al., 2024), subspace learning (Zhang et al., 2017) and graph-based clustering (Lin & Kang, 2021), typically rely on linear assumptions and handcrafted features, limiting their ability to capture complex patterns in high-dimensional data. In contrast, deep MVC methods possess the capability to model complex nonlinear relationships and high-dimensional patterns, thus attracting increasing attention. This capability has been instantiated in several representative paradigms, such as autoencoder-based frameworks (Xu et al., 2022a; Du et al., 2021), which reconstruct each view to capture rich semantic information; graph neural network-based approaches (Fan et al., 2020; Ling et al., 2023) which use graph topology as guidance to fuse attributes of each node and its neighbors, thereby generating structure-aware representations; and contrastive learning-based methods (Lin et al., 2022b; Xu et al., 2022b), which maximize mutual information across views to enforce cross-view consistency and improve cluster separability.

Despite differences in technical implementations, existing deep MVC methods consistently acknowledge that both semantic information (intrinsic sample features) and structural information (inter-sample relationships) are essential. However, most approaches emphasize one while treating the other as auxiliary, leading to imbalanced integration— for example, some prioritize constructing and utilizing consensus graphs with semantic embeddings as input (Ren et al., 2024; Du et al., 2023), while others focus on semantic reconstruction with structural information serving as guidance (Dong et al., 2025). Consequently, semantics and structure are not jointly and equitably modeled, leaving room for explicitly disentangling and adaptively fusing both sources of information. Beyond the disentanglement, the fusion of semantics and structure poses another critical challenge. Naïve feature concatenation can introduce conflicts, as features from different views, whether semantic or structural, often vary in informativeness, with certain views dominated by redundancy or noise, and their relative contributions may be dataset-dependent. Moreover, prior work typically aligns features at only one or two levels—such as cross-view consistency or intra-cluster compactness—while neglecting simultaneous multi-granularity alignment across views, within samples, and among clusters, potentially leading to suboptimal clustering and insufficient preservation of semantic-structural coherence, particularly given the heterogeneous distributions of semantic and structural embeddings.

To address these issues, we propose a Dual-branch Representations with dynamic gatEd fusion and triple-grAnularity alignMent model (DREAM), for deep MVC. Specifically, DREAM employs two dedicated encoders: a VAE encoder for semantic abstraction that captures intrinsic sample content, and a GCN encoder for structure-aware modeling that preserves inter-sample relations. Then, the extracted representations are dynamically integrated using a gated fusion module which adaptively balances semantic and structural contributions within views and leverages structural cues as complementary guidance to fuse embeddings across views, producing clustering-oriented embeddings. Finally, to further improve robustness and discriminability, we introduce a triple-granularity feature alignment mechanism that enforces consistency across views, within individual samples, and intra-cluster, thereby preserving semantic-structural coherence while enhancing inter-cluster separability. Our main contributions are summarized as follows:

- We design a dual-branch disentanglement module that explicitly separates semantic and structural information via dedicated semantics (VAE) and structure (GCN) encoders, enabling the model to capture heterogeneous information in a complementary manner.
- We propose an adaptive gated fusion module that treats semantic and structural embeddings as parallel information sources, dynamically balancing their contributions while suppressing redundant or noisy signals, thereby producing compact and discriminative latent representations.
- We introduce a unified feature alignment strategy that enforces alignment at three granularities—cross-view consistency, intra-sample coherence, and inter-cluster separability—strengthening latent feature alignment and enhancing clustering discrimination.
- Extensive experiments on multiple datasets demonstrate that DREAM outperforms SOTA methods, validating the effectiveness of the dual-branch disentanglement, adaptive fusion, and triple-granularity feature alignment mechanisms for multi-view clustering.

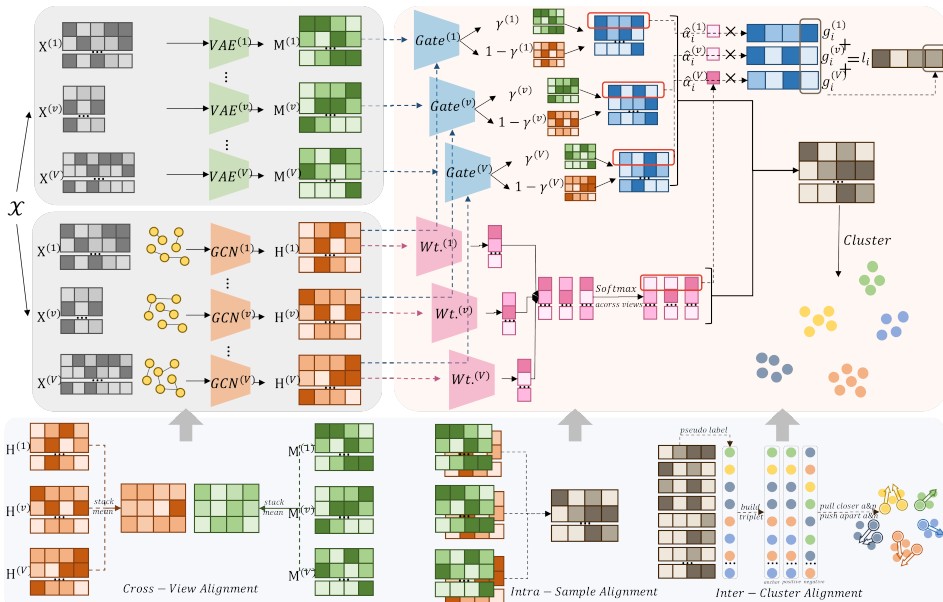

Figure 1: Framework of DREAM. Multi-view data are first encoded into semantic and structure-aware features, which are then dynamically integrated via a gated fusion module with triple-granularity alignment to produce clustering-oriented embeddings, subsequently used for clustering.

## 2 RELATED WORK

Multi-view clustering has attracted considerable attention for integrating complementary information across views, and can be broadly categorized into semantics-oriented methods, which exploit intrinsic sample attributes, and structure-oriented methods, which utilize inter-sample relationships.

**Semantics-Oriented Methods.** A common line of work in MVC emphasizes capturing rich semantic content by constructing latent embeddings, as semantics encode shared attributes that distinguish instances for clustering. For example, Wang et al. (2022) proposed a deep learning framework for MVC that factorizes view-specific data via NMF and aligns the resulting embeddings to capture consistent semantics across heterogeneous views. Lin et al. (2022a) aligned sample representations across views in a contrastive way for binary clustering tree decoding. Xu et al. (2022a) applied deep autoencoders to learn view-specific embeddings independently and concatenated them into global features to mitigate the negative impact of unclear clustering patterns in individual views. Similarly, Zeng et al. (2023) demonstrated that different views share an invariant semantic distribution, enabling the model to reduce cross-view discrepancies and learn unified semantic representations without paired samples. Li et al. (2023) proposed a dual mutual information constrained clustering method that minimizes the mutual information across all the dimensionalities to reduce the redundancy among features and maximizes the mutual information of the similar instance pairs to obtain more unbiased and robust representations. Liu et al. (2024) extracted view-specific features, integrated them according to view importance, and leveraged semantic features from both individual and fused views to generate cluster-friendly features via two dedicated contrastive losses.

**Structure-Oriented Methods.** Other approaches often prioritize capturing inter-sample structural relationships explicitly, which reveal the relative arrangement of samples and inform clustering. For instance, Xue et al. (2021) combined adaptive graph learning with graph convolution and multiple kernel clustering to integrate global and local structures for clustering. Pan & Kang (2021) filtered noisy topological information and applied graph contrastive loss to learn a consensus graph. Yan et al. (2023) aggregated features across samples and views and enforced structure-guided contrastive learning for more discriminative representations. Similarly, Wang & Feng (2024) modeled structural relations and constructed a consistent cross-view affinity matrix to enhance clustering compactness. Cui et al. (2026) leveraged local neighborhood graphs and Gaussian modeling to capture latent structural information, improving cross-view consistency and intra-cluster compactness.

In summary, semantics-oriented methods aim to achieve cross-view consistency in latent semantic spaces, while structure-oriented methods emphasize inter-sample relations via topological information. However, existing approaches often treat one type of information as primary and rely on one or two types of contrastive techniques to enforce cross-view agreement and enhance inter-class separability, leaving explicit disentanglement and adaptive balance of semantic and structural features, as well as aligning heterogeneous embedding spaces, underexplored. To address this, we propose a framework consisting of a dual-branch module for semantics and structure encoding, a gated fusion mechanism to capture heterogeneous information and to dynamically balance their contributions, and a triple-granularity contrastive objective that enforces cross-view consistency, intra-sample coherence, and inter-class separability.

## 3 METHOD

In this section, we first introduce the problem definition and provide an overview of DREAM's framework. We then describe the modules of DREAM in detail: the Dual Branch Encoding Module, the Gated Feature Fusion Module, the Feature Alignment Module, and the Clustering Module.

### 3.1 PROBLEM DEFINITION AND FRAMEWORK OVERVIEW

Let a multi-view dataset be represented as $\mathcal{X} = \{\mathbf{X}^{(1)}, \mathbf{X}^{(2)}, \ldots, \mathbf{X}^{(V)}\}$, where $V$ denotes the number of views, and $\mathbf{X}^{(v)} = \{\mathbf{x}_1^{(v)}, \mathbf{x}_2^{(v)}, \ldots, \mathbf{x}_N^{(v)}\} \in \mathbb{R}^{N \times d^{(v)}}$ represents the $N$ samples with feature dimensionality $d^{(v)}$ in the $v$-th view. The goal of multi-view clustering is to partition the $n$ samples into $K$ clusters based on their multi-view features, without access to ground-truth labels.

The overall framework of the proposed DREAM model is depicted in Figure 1. First, the multi-view dataset $\mathcal{X}$ is processed by the Semantics Encoding Branch (the upper left module with grey background) and Structure Encoding Branch (the middle left module with grey background) separately, obtaining latent feature $\mathbf{M}^{(v)}$ and $\mathbf{H}^{(v)}$. Second, for each sample $i$, latent feature $\boldsymbol{\mu}_i^{(v)}$ and $\mathbf{h}_i^{(v)}$ are processed by the Gated Feature Fusion Module (the module with light orange background) to obtain the fused latent feature $\mathbf{l}_i$. Specifically, for each view $v$, $\boldsymbol{\mu}_i^{(v)}$ and $\mathbf{h}_i^{(v)}$ are first fused via a learned gating strategy, and the resulting latent representations $\mathbf{g}_i^{(v)}$ are then aggregated across views using cross-view weighting cues derived from $\mathbf{h}_i^{(v)}$ to obtain the final fused feature $\mathbf{l}_i$. Third, $\mathbf{l}_i$ is processed by the clustering module (the bottom right part in the module with orange background), obtaining clustering result. Fourth, the module with light blue background comprises three alignment strategies corresponding to the first, second, and third step. The Cross-View Alignment module encourages each view of sample $i$ to capture more consistent semantic and structural information. The Intra-Sample Alignment module encourages the fused embedding $\mathbf{l}_i$ to remain close to its semantic and structure-aware counterparts ($\boldsymbol{\mu}_i^{(v)}$ and $\mathbf{h}_i^{(v)}$). Finally, the Inter-Cluster Alignment module enhances the discriminability among different clusters.

### 3.2 DUAL BRANCH ENCODING MODULE

Semantic and structural features are both essential for effective multi-view clustering. However, existing methods often exhibit a bias toward one type of information while treating the other as auxiliary, thereby neglecting the variability in their relative reliability across datasets as well as the inherently complementary nature of semantics and structure. Such limitations may result in reduced generalization ability and suboptimal clustering performance. To overcome these challenges, we design a Dual Branch Encoding Module that captures semantic and structural features simultaneously and separately. Specifically, the semantics encoding branch employs a variational autoencoder (VAE) to independently extract sample-level semantic content, while the structure encoding branch leverages a graph convolutional network (GCN) encoder to obtain structure-aware embeddings by explicitly modeling inter-sample relationships.

**Semantics Encoding Branch.** For each view $v$, the semantic branch employs a VAE encoder, which takes the input data $\mathbf{X}^{(v)}$ and produces the mean $\mathbf{M}^{(v)} = \{\boldsymbol{\mu}_1^{(v)}, \boldsymbol{\mu}_2^{(v)}, \ldots, \boldsymbol{\mu}_n^{(v)}\}$ and logarithm of the variance $\mathbf{S}^{(v)} = \{\log \sigma_1^{2(v)}, \log \sigma_2^{2(v)}, \ldots, \log \sigma_n^{2(v)}\}$ of the latent distribution via

$\mathbf{M}^{(v)}, \mathbf{S}^{(v)} = f_{\text{Encoder}}^{(v)}(\mathbf{X}^{(v)})$. The mean embedding $\mathbf{M}^{(v)}$ is subsequently adopted as the semantic feature representation. To ensure that $\mathbf{M}^{(v)}$ captures sufficient information from $\mathbf{X}^{(v)}$ and that the latent space follows a standard normal distribution, this branch optimizes a reconstruction loss $L_{\text{recon}}$ (Eq. 1) and a Kullback–Leibler (KL) divergence $L_{\text{KL}}^{Semantics}$ (Eq. 2):

$$L_{\text{recon}} = \sum_{v=1}^{V} \frac{1}{N} \|\hat{\mathbf{X}}^{(v)} - \mathbf{X}^{(v)}\|_2^2, \tag{1}$$

$$L_{\text{KL}}^{Semantics} = -\frac{1}{2} \sum_{v=1}^{V} \frac{1}{N} \sum_{i=1}^{N} \sum_{j=1}^{d} \left(1 + \log(\sigma_{ij}^{2(v)}) - \boldsymbol{\mu}_{ij}^{2(v)} - \sigma_{ij}^{2(v)}\right), \tag{2}$$

where $\hat{\mathbf{X}}^{(v)}$ is the reconstructed feature computed by $\hat{\mathbf{X}}^{(v)} = f_{\text{Decoder}}^{(v)}\left(\mathbf{M}^{(v)} + \exp\left(0.5\mathbf{S}^{(v)}\right) \odot \epsilon\right)$. The overall loss of the semantic branch, denoted as $L_{\text{Semantics}}$, is defined as

$$L_{\text{Semantics}} = L_{\text{recon}} + \lambda_1 L_{\text{KL}}^{Semantics}, \tag{3}$$

where $\lambda_1$ is a weighting factor that balances the reconstruction and regularization terms.

**Structure Encoding Branch.** For each view $v$, we first build the graph structure among samples, where top-k similar samples are interconnected. Please refer to Appendix A.1 for graph construction method. Then, the GCN encoder updates each sample to explicitly merge graph structure into it by $\mathbf{H}^{(v)} = \mathbf{D}^{(v)-\frac{1}{2}} \mathbf{A}^{(v)} \mathbf{D}^{(v)-\frac{1}{2}} \mathbf{X}^{(v)}$, where $\mathbf{A}^{(v)} = \{a_{ij}^{(v)}\} \in \mathbb{R}^{N \times N}$ is the adjacency matrix built in the first step, and $\mathbf{D}^{(v)}$ is the degree matrix with $d_{ii}^{(v)} = \sum_j a_{ij}^{(v)}$. To ensure that the learned embeddings preserve the original graph connectivity, we employ a graph reconstruction loss. Specifically, the predicted adjacency matrix is defined as $\hat{\mathbf{A}}^{(v)} = \sigma(\mathbf{H}^{(v)}\mathbf{H}^{(v)\top})$, where $\sigma(\cdot)$ denotes the element-wise sigmoid function. The graph reconstruction loss $L_{\text{Structure}}$ is then computed as the mean squared error between the predicted and ground-truth adjacency:

$$L_{\text{Structure}} = \sum_{v=1}^{V} \frac{1}{N^2} \|\hat{\mathbf{A}}^{(v)} - \mathbf{A}^{(v)}\|_2^2, \tag{4}$$

**Overall Encoding Loss.** Combining semantic and structural branches, the overall encoding loss is:

$$L_{\text{Encode}} = L_{\text{Semantics}} + L_{\text{Structure}}. \tag{5}$$

### 3.3 GATED FEATURE FUSION MODULE

While semantic and structural features are disentangled, fusing them remains challenging. Simple concatenation may suffer from heterogeneous feature distributions and redundancy or noise dominance. To address this, we propose a Gated Feature Fusion Module that dynamically balances semantic and structural contributions, yielding compact and discriminative embeddings. It employs Intra-View Gating to fuse two embedded features within a view, Cross-View Weighting to learn view importance and Cross-View Weighted Fusion to fuse views within each sample.

**Intra-View Gating.** Semantic and structure-aware embeddings within each view are first fused using a learnable gate:

$$\mathbf{g}_i^{(v)} = \boldsymbol{\mu}_i^{(v)} \odot \sigma\left(\mathbf{W}_{\text{Gate}}^{(v)}[\boldsymbol{\mu}_i^{(v)}\|\mathbf{h}_i^{(v)}]\right) + \mathbf{h}_i^{(v)} \odot \left(1 - \sigma(\mathbf{W}_{\text{Gate}}^{(v)}[\boldsymbol{\mu}_i^{(v)}\|\mathbf{h}_i^{(v)}])\right), \tag{6}$$

where $\boldsymbol{\mu}_i^{(v)}$ and $\mathbf{h}_i^{(v)}$ denote the semantic and structure-aware embeddings for sample $i$, $[\cdot\|\cdot]$ represents concatenation, $\mathbf{W}_{\text{Gate}}^{(v)}$ is a learnable linear projection, and $\sigma(\cdot)$ is the sigmoid activation function. This operation adaptively balances semantic and structural information within each view.

**Cross-View Weighting.** Then, for each view $v$ and sample $i$, the structure-aware embedding $\mathbf{h_i}^{(v)}$ is mapped to a scalar weight $\alpha_i^{(v)}$:

$$\alpha_i^{(v)} = f_{\text{Wt.}}^{(v)}(\mathbf{h}_i^{(v)}) \in \mathbb{R}, \tag{7}$$

where $f_{\mathrm{Wt.}}^{(v)}$ is a MLP with ReLU activation. Structure-aware embeddings $\mathbf{H}^{(v)}$ characterize sample-neighbor relations within each view, so that $\mathbf{h}_i^{(v)}$ encodes how strongly a sample connects under that view. Projecting $\mathbf{h}_i^{(v)}$ into a scalar weight $\alpha_i^{(v)}$, the model enables cross-view comparison of structural coherence and adaptively emphasizes views providing more reliable inter-sample relationship cues.

**Cross-View Weighted Fusion.** Finally, gated embeddings $\mathbf{g}_i^{(v)}$ are fused across views using the normalized weights $\hat{\alpha}_i^{(v)}$:

$$\mathbf{l}_i = \sum_{v=1}^{V} \hat{\alpha}_i^{(v)} \mathbf{g}_i^{(v)}, \quad \hat{\alpha}_i^{(v)} = \frac{\exp(\alpha_i^{(v)})}{\sum_{v'=1}^{V} \exp(\alpha_i^{(v')})}, \tag{8}$$

where $\hat{\alpha}_i^{(v)}$ is obtained via softmax normalization across views for each sample. The final fused representation $\mathbf{l}_i$ incorporates both semantic and structure-aware information, and captures complementary cues across multiple views.

### 3.4 FEATURE ALIGNMENT MODULE

In multi-view learning, decoupled and fused semantic and structure-aware features may still be inconsistent across branches and views, and fused embeddings may lose critical information or discriminability. To address this, we introduce the Feature Alignment Module, which enforces alignment at multiple levels to produce robust and informative clustering-oriented representations.

**Cross-View Alignment.** To reduce discrepancies between views, embeddings from all views are aligned toward a shared consensus using cross-view distillation losses for both the Semantics Encoding Branch and the Structure Encoding Branch:

$$L_{\mathrm{distill}}^{\mathrm{Semantics}} = \sum_{v=1}^{V} \frac{1}{N} \|\mathbf{M}^{(v)} - \mathbf{M}^*\|_2^2, \quad L_{\mathrm{distill}}^{\mathrm{Structure}} = \sum_{v=1}^{V} \frac{1}{N^2} \|\hat{\mathbf{A}}^{(v)} - \mathbf{A}^*\|_2^2, \tag{9}$$

where $\mathbf{M}^*$ and $\mathbf{A}^*$ denote the consensus targets obtained by aggregating the semantic and structure embeddings across all views. These losses encourage each view to capture consistent semantic and structural information, facilitating more coherent fused representations.

**Intra-Sample Alignment.** To preserve key semantic and structural information for each sample and maintain global discriminability across samples, a triplet-style InfoNCE loss is employed:

$$L_{\mathrm{intra}} = -\frac{1}{V} \sum_{v=1}^{V} \frac{1}{N} \sum_{i=1}^{N} \log \frac{\exp(\mathrm{sim}(\mathbf{l}_i, \boldsymbol{\mu}_i^{(v)})/\tau) + \exp(\mathrm{sim}(\mathbf{l}_i, \mathbf{h}_i^{(v)})/\tau)}{\sum_{j=1}^{N} \left[ \exp(\mathrm{sim}(\mathbf{l}_i, \boldsymbol{\mu}_j^{(v)})/\tau) + \exp(\mathrm{sim}(\mathbf{l}_i, \mathbf{h}_j^{(v)})/\tau) \right]}, \tag{10}$$

where $\mathrm{sim}(\cdot, \cdot)$ denotes cosine similarity, and $\tau$ is the temperature parameter. More specifically, for each sample $i$, the Intra-Sample Alignment loss encourages the fused embedding $\mathbf{l}_i$ to remain close to its semantic and structure-aware counterparts, i.e., $(\mathbf{l}_i, \boldsymbol{\mu}_i^{(v)})$ and $(\mathbf{l}_i, \mathbf{h}_i^{(v)})$ in the numerator, while simultaneously reduces the similarity between $\mathbf{l}_i$ and semantic and structure-aware embeddings from other samples, i.e., $(\mathbf{l}_i, \boldsymbol{\mu}_j^{(v)})$ and $(\mathbf{l}_i, \mathbf{h}_j^{(v)})$ in the denominator.

**Inter-Cluster Alignment.** To enhance the discriminability among different clusters, a triplet loss which imposes two complementary forces—an attractive force that pulls the anchor and its positives toward each other to enforce intra-cluster compactness, and a repulsive force which pushes the anchor and its negatives away from each other to ensure inter-cluster separation, is applied over the fused embeddings:

$$L_{\mathrm{inter}} = \frac{1}{R} \sum_{(a,p,n) \in \mathcal{S}} \max\left(0, \|\mathbf{l}_a - \mathbf{l}_p\|_2 - \|\mathbf{l}_a - \mathbf{l}_n\|_2 + m\right), \tag{11}$$

where $(a, p, n)$ denotes a triplet of anchor, positive, and negative samples, $m$ is the margin, $\mathcal{S}$ is the set of selected triplets, and $R$ is the total number of selected triplets. More specifically, for each triplet $(a, p, n)$, the anchor $(a)$ refers to a sample currently used as the basis to construct the triplet; the positive $(p)$ is another sample whose pseudo label is identical to that of the anchor,

meaning the two are predicted to belong to the same cluster; the negative $(n)$ is another sample whose pseudo label differs from that of the anchor. Pseudo labels are generated by the Clustering Module (Sec. 3.5) during training, where the $\arg\max_k \; p_{ik}$ is used as the pseudo label for sample $i$, updated every 3 epochs. Using pseudo labels provides dynamic refinement of cluster structure and training stability without propagating noisy signal from early-stage noisy assignments thus leading to stable convergence.

**Overall Feature Alignment Loss.** The overall feature alignment loss is formulated by integrating the cross-view, intra-sample, and inter-cluster alignment objectives:

$$L_{\text{Align}} = \lambda_2 L_{\text{distill}}^{\text{Semantics}} + \lambda_2 L_{\text{distill}}^{\text{Structure}} + L_{\text{intra}} + L_{\text{inter}}, \tag{12}$$

where $\lambda_2$ is a constant number set to 10 during experiment.

## 3.5 CLUSTERING MODULE

Finally, a clustering layer is adopted to obtain cluster assignments for fused representation $\mathbf{l}_i$. This layer maintains a set of trainable cluster centers $\{\boldsymbol{c}_k\}_{k=1}^K$ where $K$ denotes the number of clusters.

The similarity between latent representation $\mathbf{l}_i$ and each cluster center is first measured using a Student's $t$-distribution kernel $q_{ik} = \frac{(1+\|\mathbf{l}_i-\boldsymbol{c}_k\|_2^2)^{-1}}{\sum_{j=1}^K (1+\|\mathbf{l}_i-\boldsymbol{c}_j\|_2^2)^{-1}}$ to provide a soft assignment of sample $i$ to each cluster, where $q_{ik}$ represents soft assignment distribution of sample $i$ to cluster $k$. Before computing distances, both $\mathbf{l}_i$ and $\boldsymbol{c}_k$ are $l_2$-normalized to improve numerical stability. Then, a sharpened target distribution $\mathbf{p}_i$ is generated through $p_{ik} = \frac{q_{ik}^{1/tem}}{\sum_{j=1}^K q_{ij}^{1/tem}}$, where $tem$ is the temperature parameter controlling the sharpness of the distribution. Finally, each sample is assigned to the cluster with the highest probability $y_{\text{pred}}^{(i)} = \arg\max_k \; p_{ik}$.

During this process, an entropy loss $L_{\text{entropy}}$ (Eq. 13) encouraging each sample's soft assignment to be confident (i.e., close to one-hot), and a KL divergence $L_{\text{KL}}^{Cluster}$ (Eq. 14) aligning the predicted soft assignments $q$ with a more confident target distribution $p$ to enhance cluster purity, are leveraged.

$$L_{\text{entropy}} = -\frac{1}{N} \sum_{i=1}^N \sum_{k=1}^K q_{ik} \log(q_{ik} + \epsilon), \tag{13}$$

$$L_{\text{KL}}^{Cluster} = \text{KL}(p\|q) = \frac{1}{N} \sum_{i=1}^N \sum_{k=1}^K p_{ik} \log \frac{p_{ik}}{q_{ik} + \epsilon}, \tag{14}$$

where $\epsilon$ is a small constant to avoid numerical issues.

**Overall Cluster Loss.** The overall cluster loss is:

$$L_{\text{Cluster}} = L_{\text{entropy}} + \lambda_3 L_{\text{KL}}^{Cluster}, \tag{15}$$

where $\lambda_3$ is used to balance the two losses.

## 3.6 THE OVERALL OPTIMIZATION OBJECTIVE

By jointly considering the Dual Branch Encoding Module, the Gated Feature Fusion Module, the Feature Alignment Module, and the Clustering Module, the overall objective function of the DREAM model can be formulated as:

$$L_{\text{Total}} = L_{\text{Encode}} + \alpha L_{\text{Align}} + \beta L_{\text{Cluster}}, \tag{16}$$

where $\alpha$ and $\beta$ are hyperparameters that adjust the contributions of three losses. To make a clear presentation, the algorithm flow is shown in Algorithm 1.

# 4 EXPERIMENT

## 4.1 EXPERIMENTAL SETTINGS

**Datasets:** Six widely used multi-view datasets—Yale, NGS, BBC, UCI, HW, and ALOI100—are employed for clustering experiments. Details of these datasets are summarized in Table 1.

---

**Algorithm 1** The optimization of DREAM

---

**Require:** Multi-view dataset $\{\mathbf{X}^{(v)}\}_{v=1}^V$; Cluster number $K$
 1: Initialize graph structures $\{\mathbf{A}^{(v)}\}_{v=1}^V$ and cluster centers $\{\boldsymbol{c}_k\}_{k=1}^K$
 2: **while** Not reach the maximum iteration $T_{\max}$ **do**
 3:     **for** each view $v = 1$ to $V$ **do**
 4:         Extract semantic features $\mathbf{M}^{(v)}$ and structure features $\mathbf{H}^{(v)}$ (Sec. 3.2)
 5:         Calculate Encoding Loss (Eq. 5)
 6:     **end for**
 7:     Fuse $\mathbf{M}^{(v)}$ and $\mathbf{H}^{(v)}$ (Sec. 3.3), and calculate Feature Alignment Loss (Sec. 3.4, Eq. 12)
 8:     Perform clustering (Sec. 3.5) and calculate Cluster Loss (Sec. 3.5, Eq. 15)
 9:     Jointly optimize the overall objective function $L_{\text{Total}}$ (Eq. 16)
10:     Backpropagate and update model parameters
11: **end while**
12: **Return:** The final clustering result

---

**Comparison Methods:** Eight representative SOTA MVC methods are used for comparison, including DSMVC (Tang & Liu, 2022), MFLVC (Xu et al., 2022b), SEM (Xu et al., 2023), GCFAg-gMVC (Yan et al., 2023), SCMVC (Wu et al., 2024), MVCAN (Xu et al., 2024), SCM (Luo et al., 2024), and GDMVC (Bai et al., 2024).

**Evaluation Metrics:** Clustering performance is evaluated using three standard metrics: clustering accuracy (ACC), normalized mutual information (NMI), and purity. Higher metric values indicate better performance.

**Implementation Details:** Our model is implemented in PyTorch 2.7.1 and trained on a desktop equipped with an NVIDIA GeForce RTX 5070 Ti GPU and 64 GB of RAM, using the Adam optimizer with default settings. Given the variations in sample sizes, feature dimensions, and the number of views across datasets, hyperparameters are tuned for each dataset from a candidate range to obtain optimal configurations. The learning rate used for different datasets ranges between $[0.1, 0.00005]$. For baseline methods, we use hyperparameters recommended in their papers or released codes and perform a light search around these defaults to prevent performance loss due to mismatched settings and report the best result.

Table 1: Summary of datasets used for clustering experiments.

| Dataset | Samples | Views | Clusters |
|---|---|---|---|
| Yale | 165 | 3 | 15 |
| NGS | 500 | 3 | 5 |
| BBC | 685 | 4 | 5 |
| UCI | 2000 | 3 | 10 |
| HW | 2000 | 6 | 10 |
| ALOI100 | 10800 | 4 | 100 |

## 4.2 COMPARISON RESULTS

Table 2 reports the experimental performance of our DREAM model and eight baseline methods across different datasets. The best performance for each metric is highlighted in **bold**, while the second-best is indicated with underlining. As shown in Table 2, different methods exhibit varying performance across datasets. Our method, DREAM, consistently outperforms all baselines on six benchmark datasets and three evaluation metrics, demonstrating its strong generalization ability and robustness. For instance, on the ALOI100 dataset, DREAM surpasses the second-best method, GDMVC, by 5.19%, 4.22%, and 5.93% in ACC, NMI, and Purity, respectively. This improvement clearly validates the effectiveness of the disentangled dual-branch encoding, the adaptive gated fusion and the triple-granularity alignment in enhancing multi-view clustering performance. Please see further discussion in Appendix A.7.

## 4.3 ABLATION STUDIES

To clearly illustrate the contribution of each core component in DREAM, we conduct a systematic ablation study by removing the Semantics Encoding Module, Structure Encoding Module, Gated Feature Fusion Module, and Feature Alignment Module individually. Table 3 reports the performance under each ablation setting.

First, the two encoding branches are examined. Removing the Semantics Encoding Module leads to a consistent drop in performance across datasets, indicating that semantic representations provide

Table 2: Performance comparison of multi-view clustering algorithms on six benchmark datasets.

| Datasets | DSMVC | MFLVC | SEM | GCFAggMVC | SCMVC | MVCAN | SCM | GDMVC | Ours |
|---|---|---|---|---|---|---|---|---|---|
| | | | | **ACC** | | | | | |
| Yale | 64.85 | 21.82 | 27.27 | 30.91 | 39.39 | 40.60 | 49.09 | 76.97 | **78.18** |
| NGS | 40.00 | 90.40 | 93.80 | 88.60 | 93.20 | 30.60 | 97.20 | 40.20 | **97.80** |
| BBC | 42.48 | 77.81 | 61.46 | 58.98 | 86.57 | 78.54 | 58.98 | 34.89 | **90.07** |
| UCI | 93.75 | 86.05 | 87.05 | 83.20 | 68.65 | 92.00 | 67.45 | 85.50 | **95.90** |
| HW | 95.85 | 68.40 | 81.35 | 81.25 | 82.40 | 95.10 | 84.25 | 88.25 | **97.80** |
| ALOI100 | 15.66 | 7.06 | 65.81 | 4.85 | 4.34 | 67.98 | 3.96 | 81.81 | **87.00** |
| | | | | **NMI** | | | | | |
| Yale | 66.81 | 19.48 | 35.11 | 32.84 | 40.37 | 45.48 | 51.92 | 76.79 | **79.87** |
| NGS | 12.26 | 76.05 | 81.89 | 74.70 | 82.20 | 17.51 | 91.03 | 12.19 | **92.90** |
| BBC | 11.26 | 59.20 | 44.12 | 53.26 | 71.53 | 63.26 | 31.39 | 4.85 | **72.75** |
| UCI | 89.24 | 79.00 | 76.72 | 72.95 | 65.88 | 85.23 | 60.20 | 86.15 | **92.01** |
| HW | 92.21 | 66.68 | 71.58 | 72.67 | 73.53 | 89.75 | 73.41 | 89.37 | **95.05** |
| ALOI100 | 40.69 | 37.90 | 82.97 | 14.80 | 12.47 | 83.76 | 11.47 | 86.66 | **90.88** |
| | | | | **Purity** | | | | | |
| Yale | 65.45 | 21.82 | 27.88 | 5.44 | 40.61 | 43.03 | 52.12 | 76.97 | **82.42** |
| NGS | 41.40 | 90.40 | 93.80 | 88.60 | 93.20 | 33.40 | 97.20 | 40.60 | **98.00** |
| BBC | 44.96 | 77.81 | 65.99 | 68.91 | 86.57 | 78.54 | 58.98 | 37.81 | **90.07** |
| UCI | 93.75 | 86.05 | 87.05 | 83.20 | 72.10 | 92.00 | 68.80 | 85.50 | **96.30** |
| HW | 95.85 | 68.55 | 81.35 | 81.25 | 82.40 | 95.10 | 84.25 | 88.25 | **97.80** |
| ALOI100 | 16.36 | 7.06 | 68.61 | 5.20 | 4.51 | 72.17 | 4.11 | 82.25 | **88.18** |

Table 3: Ablation studies on the contributions of each component in the DREAM model.

| Datasets | UCI | | | HW | | | ALOI100 | | |
|---|---|---|---|---|---|---|---|---|---|
| Metrics | ACC | NMI | Purity | ACC | NMI | Purity | ACC | NMI | Purity |
| w/o Semantics Encoding | 87.05 | 80.44 | 87.05 | 90.55 | 87.00 | 92.05 | 84.68 | 89.24 | 86.69 |
| w/o Structure Encoding | 75.90 | 67.85 | 78.40 | 84.65 | 86.30 | 94.50 | 78.29 | 87.32 | 82.16 |
| w/o Gated Fusion | 82.70 | 83.44 | 85.70 | 92.85 | 87.82 | 94.00 | 83.90 | 89.73 | 85.07 |
| w/o Feature Alignment | 88.35 | 81.58 | 89.75 | 96.25 | 91.83 | 96.25 | 86.62 | 90.61 | 87.84 |
| Our model | **95.90** | **92.01** | **96.30** | **97.80** | **95.05** | **97.80** | **87.00** | **90.88** | **88.18** |

indispensable, discriminative cues for clustering. Removing the Structure Encoding Module results in the most severe performance degradation—for example, ACC decreases by 20% on the UCI dataset—demonstrating that inter-sample structural relations are fundamental for reliable multi-view clustering. Next, the importance of the fusion mechanism is assessed by replacing the Gated Feature Fusion Module with simple averaging. While the changed model occasionally surpasses single-branch variants, it remains substantially inferior to the full DREAM model. This indicates that naive averaging fails to effectively leverage the complementary information of semantic and structural embeddings, whereas gated fusion adaptively balances view informativeness. Finally, disabling the Feature Alignment Module also results in noticeable performance degradation. On the UCI dataset, ACC decreases from 95.90% to 88.35%, demonstrating that triple-granularity alignment strengthens cluster cohesion by harmonizing representations across views, samples, and clusters.

Overall, the ablation results validate that each module contributes meaningfully and that the integration of dual-branch encoding, gated feature fusion, and feature alignment achieves superior multi-view clustering performance.

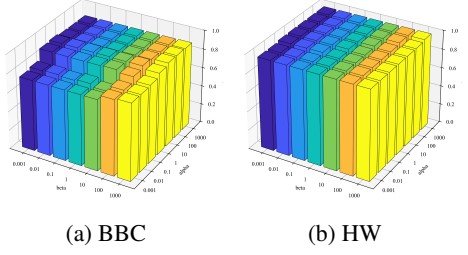

(a) BBC      (b) HW

## 4.4 SENSITIVITY ANALYSIS

We conducted a sensitivity analysis on the hyperparameters of DREAM (Figure 2). Specifically, we investigated two key hyperparameters, $\alpha$ and $\beta$, by varying their values across the range

Figure 2: Impact of varying the hyperparameters $\alpha$ and $\beta$ on clustering performance.

$[0.001, 0.01, 0.1, 1, 10, 100, 1000]$. Results show that changes in these parameters induce only minor fluctuations in performance on the BBC and HW datasets, indicating that our method is highly robust to hyperparameter selection. Additional results on other datasets are provided in Appendix A.5.

## 4.5 CONVERGENCE ANALYSIS

We ploted metrics and losses over training iterations with average (avg) and standard deviation (std) from five random-seed experiments on BBC and HW (Figure 3) to demonstrate the robustness of our model. Results show that metrics and losses stabilize with the training cycle, indicating that the model shows good convergence properties.

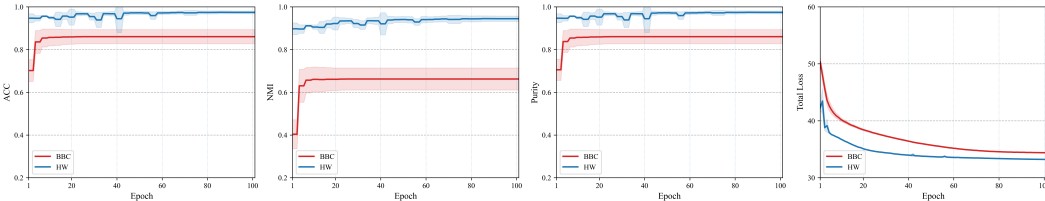

Figure 3: Convergence analysis on BBC and HW.

## 4.6 VISUALIZATION

To qualitatively assess the clustering capability of DREAM, we performed t-SNE visualizations on both the raw features and the fused features learned by DREAM on the BBC and HW datasets (Figure 4). Compared with the raw features, the learned fused features exhibit a markedly clearer separation of clusters, indicating that DREAM effectively captures highly discriminative representations that are well-suited for clustering tasks.

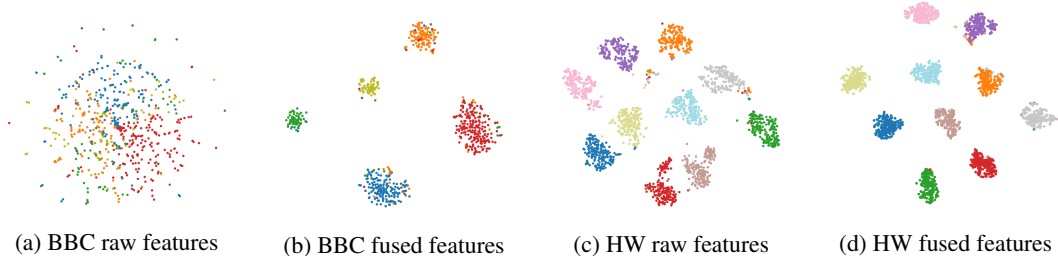

(a) BBC raw features    (b) BBC fused features    (c) HW raw features    (d) HW fused features

Figure 4: t-SNE visualization of raw and learned fused features on BBC and HW datasets.

## 5 CONCLUSION

In this work, we present DREAM, a novel multi-view clustering framework designed to disentangle and integrate semantic and structural information to improve clustering performance. DREAM introduces three innovative components: a dual-branch encoder that separately models semantic and structure-aware representations, a gated fusion module that adaptively balances contributions of representations, and a triple-granularity alignment strategy that enforces consistency across views, within individual samples, and within clusters. Comprehensive experiments on multiple benchmark datasets demonstrate that DREAM consistently surpasses SOTA methods, highlighting its effectiveness and generality for multi-view clustering.

## 6 REPRODUCIBILITY STATEMENT

We have made efforts to ensure the reproducibility of our results. The details of method are provided in Sec. 3, and details of model architecture are described in Appendix. A.6. The code of our model will be made publicly available upon publication.

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

# A APPENDIX

## A.1 GRAPH CONSTRUCTION

Two graph structure initialization methods are leveraged in our experiment for datasets of different kind. For image datasets, including ALOI100, HW, Yale and UCI, k-nearest neighbors are obtained by computing Euclidean distance to construct graphs for each view feature matrix $\mathbf{X}^{(v)}$. This approach is efficient for dense image features and leverages geometric distance which is natural for visual descriptors. For text datasets, including NGS and BBC, a cosine-similarity-based k-nearest graph is built. Specifically, first, each view feature $\mathbf{X}^{(v)}$ is L2-normalized; next, the cosine similarity matrix is computed and its diagonal is set to zero to avoid self-loops; finally, the top-k largest similarity entries are kept as neighbors, producing a graph. This procedure is robust for sparse while high-dimensional textual representations and explicitly controls local connectivity.

Table 4: Comparison between cosine-similarity and Euclidean-distance graph construction methods on HW and BBC. For brevity, COS and EUC are used in the table to represent cosine-similarity and Euclidean-distance, respectively.

| Datasets | ACC | | NMI | | Purity | |
|---|---|---|---|---|---|---|
| | COS | EUC | COS | EUC | COS | EUC |
| HW | 97.50 | 97.80 | 94.73 | 95.05 | 97.50 | 97.80 |
| BBC | 90.07 | 24.23 | 72.75 | 1.31 | 90.07 | 33.87 |

To further demonstrate the influence of graph construction strategies on model performance, we conducted an experiment comparing cosine-similarity graphs and Euclidean-distance graphs under the same hyperparameter settings on two representative datasets HW and BBC. The results are summarized in Table 4. It can be observed that while modifying the graph construction strategy for HW leads to only a slight performance drop, applying the same change to BBC results in a substantial degradation. This indicates that text-based datasets, such as BBC, are highly sensitive to the graph construction method and particularly benefit from cosine-similarity-based graph construction method.

## A.2 RATIONALE FOR USING GCN ENCODER RATHER THAN GROUND-TRUTH ADJACENCY

To verify the necessity of the GCN encoder, we conducted an experiment in which the GCN encoder is replaced by a linear layer that directly receives the ground-truth adjacency matrix as input. The linear layer is employed because the next module requires inputs of the same size. The results, summarized in Table 5, show that while using the ground-truth adjacency achieves reasonable (good though not better) performance on image datasets (Yale, UCI, HW, and ALOI100), it leads to a significant performance drop on text-based datasets (NGS and BBC). These findings demonstrate that the GCN encoder is essential for effectively dealing with both image- and text-based datasets simultaneously.

Table 5: Comparison between using GCN encoder and Ground-truth Adjacency.

|  | Yale | NGS | BBC | UCI | HW | ALOI100 |
|---|---|---|---|---|---|---|
| **ACC** | | | | | | |
| Ours (with GCN encoder) | 78.18 | 97.80 | 90.07 | 95.90 | 97.80 | 87.00 |
| Ours (with Ground-truth Adjacency) | 77.58 | 56.60 | 51.24 | 92.90 | 94.50 | 81.19 |
| **NMI** | | | | | | |
| Ours (with GCN encoder) | 79.87 | 92.90 | 72.75 | 92.01 | 95.05 | 90.88 |
| Ours (with Ground-truth Adjacency) | 78.28 | 31.93 | 27.00 | 89.56 | 92.31 | 89.03 |
| **Purity** | | | | | | |
| Ours (with GCN encoder) | 82.42 | 98.00 | 90.07 | 96.30 | 97.80 | 88.18 |
| Ours (with Ground-truth Adjacency) | 80.61 | 61.20 | 59.56 | 95.25 | 97.15 | 84.85 |

## A.3 RATIONALE FOR USING STRUCTURAL-CUES IN THE GATED FEATURE FUSION MODULE

The rationale for using structure-aware embedding $\mathbf{H}^{(v)}$ as guidance for cross-view weighting is twofold. First, as stated in Sec. 3.3, the structure-aware embedding $\mathbf{h}_i^{(v)}$ incorporates inter-instance information, thus hints the structural reliability of sample $i$ in view $v$. In other words, it captures how consistently this sample aligns with its local neighborhood in that view's graph structure. This property provides valuable cues for view weighting during fusion. Second, an experiment replacing the structural cues with semantic cues is conducted to verify the effectiveness of structural cue-aided gated feature fusion (Table 6).

Table 6: Comparison between using structural cues and semantic cues in the Gated Feature Fusion Module.

|  | Yale | NGS | BBC | UCI | HW | ALOI100 |
|---|---|---|---|---|---|---|
| **ACC** | | | | | | |
| Ours (with structural-guided fusion) | 78.18 | 97.80 | 90.07 | 95.90 | 97.80 | 87.00 |
| Ours (with semantic-guided fusion) | 76.97 | 86.00 | 87.74 | 93.85 | 97.55 | 85.99 |
| **NMI** | | | | | | |
| Ours (with structural-guided fusion) | 79.87 | 92.90 | 72.75 | 92.01 | 95.05 | 90.88 |
| Ours (with semantic-guided fusion) | 76.60 | 73.04 | 68.25 | 89.75 | 94.39 | 90.86 |
| **Purity** | | | | | | |
| Ours (with structural-guided fusion) | 82.42 | 98.00 | 90.07 | 96.30 | 97.80 | 88.18 |
| Ours (with semantic-guided fusion) | 80.00 | 88.40 | 87.74 | 95.10 | 97.55 | 89.56 |

## A.4 ABLATION STUDIES

Experiment reuslts on NGS, Yale and BBC datasets are reported in Table 7, from which we can see that all the modules in our model are verified as indispensable. To be specific, removing either the semantic or structural encoding module leads to clear performance degradation, with the structural branch being especially critical. Disabling the gated fusion module also reduces accuracy, confirming that simple averaging cannot fully exploit semantic-structural complementarity. Finally, the feature alignment module is essential for maintaining representation consistency, as its removal noticeably lowers performance.

Interestingly, the BBC dataset shows different trends from NGS and Yale. When Semantics Encoding Module is removed, the model achieves even higher accuracy than the full model version. This is likely because BBC is inherently a text-based dataset whose raw views already carry strong and highly correlated semantic signals. Adding an explicit Semantics Encoding Module may rather introduce redundancy or noise, leading to inferior performance. Instead, the scarce structural information in the raw data becomes relatively more valuable. These results highlight that the relative

importance of semantic and structural information is dataset-dependent, further validating the necessity of our disentangled design.

Table 7: Ablation studies on NGS, Yale and BBC datasets.

| Datasets | NGS | | | Yale | | | BBC | | |
|---|---|---|---|---|---|---|---|---|---|
| Metrics | ACC | NMI | Purity | ACC | NMI | Purity | ACC | NMI | Purity |
| w/o Semantics Encoding | 95.20 | 87.56 | 95.20 | 72.12 | 72.82 | 76.36 | **91.53** | **76.72** | **91.53** |
| w/o Structure Encoding | 45.60 | 19.74 | 45.60 | 67.88 | 68.37 | 69.09 | 29.20 | 3.48 | 35.33 |
| w/o Gated Feature Fusion | 96.60 | 91.18 | 97.00 | 73.94 | 73.45 | 76.97 | 82.34 | 63.77 | 83.65 |
| w/o Feature Alignment | 62.40 | 50.08 | 68.80 | 74.55 | 77.63 | 81.21 | 75.62 | 47.42 | 75.62 |
| Our model | **97.80** | **92.90** | **98.00** | **78.18** | **79.87** | **82.42** | 90.07 | 72.75 | 90.07 |

## A.5 SENSITIVITY STUDIES

Two hyperparameters, $\alpha$ and $\beta$, are varied across the range $[0.001, 0.01, 0.1, 1, 10, 100, 1000]$ to examine their impact on model performance. Results (Figure 5) show that varying these parameters causes only minor performance variations across the studied datasets, confirming the robustness of our approach.

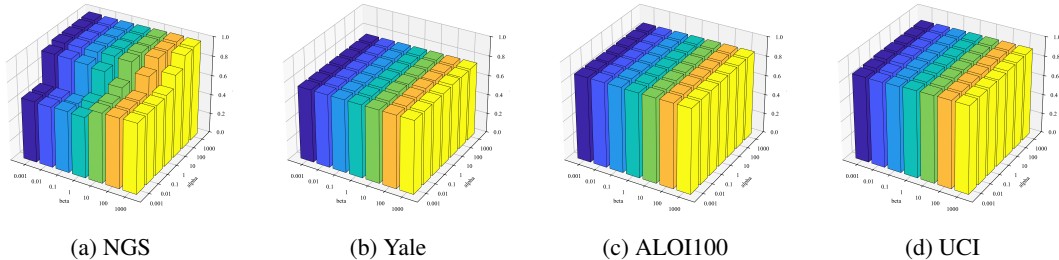

| (a) NGS | (b) Yale | (c) ALOI100 | (d) UCI |
|---|---|---|---|

Figure 5: Sensitivity analysis of hyperparameters $\alpha$ and $\beta$ on four datasets.

## A.6 DETAILS OF MODEL ARCHITECTURE

**Dual Branch Encoding Module.** The semantic encoder in our model consists of three linear layers with ReLU activations and an additional linear layer yielding the mean and log-variance for the latent semantic representation. The structural encoder is composed of three graph convolutional layers with ReLU activations.

**Gated Feature Fusion Module.** It leverages one linear layer with sigmoid activation to adaptively fuse embeddings from two encoders within each view, and another two linear layers with ReLU activation in between to learn view importance.

**Feature Alignment Module.** This module mainly use loss functions defined in Sec. 3.4 to align features at three granularities. It contains no trainable layers.

**Clustering Module.** This module initializes trainable cluster centers using K-means on the fused features, and iteratively refines them via soft assignment.

## A.7 FURTHER COMPARISON BETWEEN DREAM AND EXISTING DEEP LEARNING-BASED APPROACHES

While existing methods may struggle to achieve satisfactory clustering results for small-scale datasets (e.g., Yale, BBC) and datasets with a large number of fine-grained classes (e.g., ALOI100), DREAM achieves a significant clustering improvement for these datasets. We believe that the difficulty faced by existing methods on these tasks primarily stems from representation unreliability and insufficient class-specific information. When only a small number of samples are available per class, it is difficult for the model to extract enough discriminative cues, and the learned embeddings may contain misleading signals or lack informative structure, ultimately leading to suboptimal clustering

performance. Our method achieves improvements on these challenging datasets for the following two reasons. First, dual-branch disentanglement with adaptive feature fusion enhances information richness and representation reliability. Unlike methods that extract a single type of information and embed it into one latent space, our framework explicitly disentangles semantic information (via the VAE branch) and structural information (via the GCN branch), and integrates them through a gated fusion mechanism. This design not only compels the model to learn complementary perspectives of the data, thereby enriching information diversity, but also enables it to dynamically balance the semantic and structural contributions according to the informativeness of each, thereby improving the reliability of the learned representations. Second, triplet-alignment improves robustness against fine-grained noise. Datasets such as ALOI100 contain fine-grained categories, leading to subtle inter-class differences. Consequently, representations are more susceptible to noise. Our alignment mechanism jointly aligns latent spaces across views, across information types, and within clusters, forcing the model to aggregate information from multiple sources, maintaining consistent representations while mitigating noise.

## A.8 THE USE OF LARGE LANGUAGE MODELS (LLMS)

The language of this manuscript was refined with the assistance of a large language model (LLM). The LLM was also consulted during the early idea-formation stage to assist in reviewing relevant literature. All other parts of this paper, including experiments, analyses, and conclusions, were designed and conducted solely by the authors.

