# OpenReview forum: "Dual-Branch Representations with Dynamic Gated Fusion and Triple-Granularity Alignment for Deep Multi-View Clustering"
_ICLR.cc/2026/Conference — ICLR 2026 Poster_

### Official Review · Reviewer_K17d · 2025-10-28

**Soundness:** 3
**Presentation:** 3
**Contribution:** 3
**Rating:** 6
**Confidence:** 5

**Summary:**

The paper proposes DREAM, a deep multi-view clustering framework that jointly exploits semantic and structural information. DREAM disentangles semantic features via a VAE branch and captures structure-aware representations via a GCN branch, dynamically integrating them through a gated fusion mechanism that adaptively balances their contributions. Moreover, a triple-granularity alignment strategy enforces consistency across views, samples, and clusters, enhancing robustness and discriminability.

**Strengths:**

1. The paper proposes a dual-branch disentanglement module that explicitly separates semantic and structural information via dedicated encoders—a VAE for semantics and a GCN for structure—allowing the model to capture heterogeneous information in a complementary manner. This is an interesting and potentially valuable idea.
2. Experimental results on six benchmark datasets demonstrate that the proposed method outperforms other approaches.
3. The ablation study verifies the effectiveness of each module in the proposed model.

**Weaknesses:**

1. Some parts of the model description are unclear; for example, the value of $\lambda_1$ in Equation (3) is not provided.
2. The Feature Alignment Module is one of the innovations of the paper; however, the description of the Overall Feature Alignment Loss is not sufficiently clear. In Equation (10), both $L_{\text{Semantic}}$ and $L_{\text{Structure}}$ share the same hyperparameter $\lambda_2$, but it is unclear why they are set equal. In addition, $L_{\text{intra}}$ and $L_{\text{inter}}$ do not have any hyperparameters, and the rationale for this choice is not explained.
3. The manuscript refers to a "Cross-View Weighting" mechanism within the Gated Feature Fusion Module. Nevertheless, it is not evident how the cross-view component is realized, as Equation (7) does not explicitly reflect any cross-view fusion process. The authors are encouraged to clarify this implementation detail.
4. The manuscript contains formatting issues; for example, a line break occurs at lines 458--459.

**Questions:**

See Weakness section.

---

> ### Author Response · Authors · 2025-11-19
>
> We are grateful to Reviewer K17d for the insightful and valuable feedback. We have thoroughly considered each comment and prepared the clarification accordingly. We hope that our responses satisfactorily resolve all of the reviewer’s concerns.
>
> **Q1. Some parts of the model description are unclear; for example, the value of $\lambda_1$ in Equation (3) is not provided.**
>
> A1: Thank you for pointing out this. Values of $\lambda_1$ are not shared across datasets so that we didn’t provide an exact value in our paper. This hyperparameter is designed to regulate the relative contribution of the KL divergence. In our initial experiments, we observed that the KL term is sensitive to dataset characteristics and can noticeably affect the clustering performance. Therefore, $\lambda_1$ is selected from a candidate range via experiments to obtain a better performance for each dataset.
>
> **Q2. The Feature Alignment Module is one of the innovations of the paper; however, the description of the Overall Feature Alignment Loss is not sufficiently clear. In Equation (10), both $ L_\mathrm{Semantic}$ and $ L_\mathrm{Structure}$ share the same hyperparameter $\lambda_2$, but it is unclear why they are set equal. In addition, $L_\mathrm{intra}$ and $L_\mathrm{inter}$ do not have any hyperparameters, and the rationale for this choice is not explained.**
>
> A2: Thank you for this valuable comment. The Overall Feature Alignment Loss integrates $L_\mathrm{distill}^{\mathrm{Semantics}}$, $L_\mathrm{distill}^{\mathrm{Structure}}$, $L_\mathrm{intra}$ and $L_\mathrm{inter}$ in Equation (12) and a single balancing coefficient $\lambda_2$  (fixed to 10) is used to regulate the contribution of these terms.
>
> Reason for using the same $\lambda_2$ for $L_\mathrm{distill}^{\mathrm{Semantics}}$ and $L_\mathrm{distill}^{\mathrm{Structure}}$. These two losses are symmetric view-level objectives. Both distillation terms aim to extract consistent information across multiple views of the same underlying source (semantic or structural), ensuring that each sample can obtain a shared and view-consensus representation. Since they serve an equivalent role but operate on different modalities, coupling them under the same weight avoids introducing unnecessary hyperparameters and keeps the optimization stable.
>
> Reason for not assigning additional hyperparameters to$L_\mathrm{intra}$ and $L_\mathrm{inter}$. $L_\mathrm{intra}$ and $L_\mathrm{inter}$ are both sample-level alignment objectives, jointly aiming to enhance the discriminability of latent embeddings. Since these two losses operate on the same level of representation and contribute to the same goal, it is natural to treat them under a unified weighting scheme. Moreover, our preliminary experiments indicate that adding extra hyperparameters to $L_\mathrm{intra}$ and $L_\mathrm{inter}$ does not improve performance noticeably while increasing tuning complexity. We therefore adopt the more compact formulation without additional coefficients.
>
> This design ensures that view-level consistency and sample-level discriminability are jointly optimized with minimal hyperparameter overhead, while maintaining effective training across datasets.

---

> ### Author Response · Authors · 2025-11-19
>
> **Q3. The manuscript refers to a "Cross-View Weighting" mechanism within the Gated Feature Fusion Module. Nevertheless, it is not evident how the cross-view component is realized, as Equation (7) does not explicitly reflect any cross-view fusion process. The authors are encouraged to clarify this implementation detail.**
>
> A3: Thank you for the meaningful comment. In Equation (7), we use a MLP layer to map structure-aware embedding $\mathbf{h}_i^{(v)}$ into a scalar weight $\alpha_i^{(v)}$, which hints the reliability of view $v$ when all the views are being fused. Then, in Equation (8), $\alpha_i^{(v)}$ is normalized by softmax across all views for each sample to represent the weight for each view. The normalized weight $\hat{\alpha}_i^{(v)}$ is used to multiple the gated feature $\mathbf{g}_i^{(v)}$, which is obtained through Intra-View Gating by combining information from semantic and structure-aware embeddings, to realize the fusion among all views.
>
> **Q4. The manuscript contains formatting issues; for example, a line break occurs at lines 458--459.**
>
> A4: Thank you for raising this point. The line break exists because the caption of Figure 2 occupies some space of line 459. We will refine it in our final version.

---

### Official Review · Reviewer_yNg5 · 2025-10-28

**Soundness:** 3
**Presentation:** 3
**Contribution:** 3
**Rating:** 6
**Confidence:** 5

**Summary:**

The manuscript proposes a multi-view clustering method that explicitly leverages both semantic and structural representations. The approach first disentangles semantic and structure-aware features using a dual-branch architecture and then adaptively integrates them through a gated fusion mechanism. To further enhance robustness and inter-cluster separability, a triple-granularity feature alignment strategy is applied across views, samples, and clusters. The manuscript also presents extensive experimental results demonstrating the effectiveness of the proposed method.

**Strengths:**

1. The manuscript is well written, and the overall structure is logically organized.

2. The proposed method effectively enhances performance by jointly exploring multi-view semantic and structural information, performing cross-view fusion, and introducing multi-level alignment strategies to fully exploit the rich information across views.

3. The experimental section includes both performance comparisons and visualization analyses, which effectively validate the effectiveness of the proposed approach.

**Weaknesses:**

1. The flowchart suggests that pseudo-label information is incorporated into the framework. However, the manuscript does not clearly explain how these pseudo labels are generated or what specific role they play in the learning process. The authors should clarify the source of the pseudo labels and elaborate on how they contribute to model optimization and clustering performance.

2. In the Inter-Cluster Alignment loss, the roles of (a, p, n) are not sufficiently clarified. Their specific definitions and underlying physical interpretations should be explicitly described.

3. Apart from the balancing parameter in the model, the descriptions of other hyperparameter settings are insufficient. It remains unclear whether these parameters are kept consistent across datasets, how their values are determined, and whether the comparisons with baseline methods are conducted under fair and comparable conditions.

4. The experimental section would benefit from a convergence analysis.

**Questions:**

See Weakness.

---

> ### Author Response · Authors · 2025-11-19
>
> We would like to express our sincere appreciation to Reviewer yNg5 for the thorough review and constructive comments. We have carefully prepared detailed responses to address each concern. We hope that the clarifications fully resolve the reviewer’s concerns.
>
> **Q1. The flowchart suggests that pseudo-label information is incorporated into the framework. However, the manuscript does not clearly explain how these pseudo labels are generated or what specific role they play in the learning process. The authors should clarify the source of the pseudo labels and elaborate on how they contribute to model optimization and clustering performance.**
>
> A1: We thank the reviewer for the valuable comment. Our framework indeed incorporates pseudo-label information, and we clarify the details below.
>
> Pseudo-label generation. In our method, pseudo labels are generated directly by the Clustering Module during training. Specifically, at each epoch, the fused embedding $\mathbf{l}_i$ is passed to the Clustering Module to produce a soft assignment distribution $q$. Every three epochs, we update the target distribution $p$ by applying a sharpening function to $q$, which enhances assignment confidence while suppressing early-stage noise amplification. The pseudo labels $\hat{y}$ are then obtained by taking the argmax of $p$. These labels are generated dynamically throughout training and are not tied to any external supervision.
>
> Role of pseudo labels. These pseudo labels are used exclusively in the Inter-Cluster Alignment Module. They serve as the guidance for the construction of triplets $\left(a,p,n \right)$ ($a$ represents anchor sample, $p$ represents positive sample, and $n$ represents negative sample), where samples with the same pseudo label serve as positives, and samples with different pseudo labels serve as negatives. The Inter-Cluster Alignment loss (Equation (11)) encourages intra-cluster compactness and inter-cluster separation based on the current clustering structure inferred by the model.
>
> Benefits to the learning process. The pseudo-label mechanism provides two key benefits to our framework:
> 1. Dynamic refinement of cluster structure. The iterative sharpening and triplet-based alignment progressively refine cluster boundaries, improving global discriminability of the fused representations.
> 2. Stability without propagating noisy signals. Updating $p$ every three epochs (instead of every iteration) prevents oscillation from early-stage noisy assignments and leads to stable convergence.
>
> Importantly, pseudo labels are not used as ground-truth labels and thus do not introduce any form of unintended supervision. They solely serve as an internal self-supervised signal to enhance cluster separation.
>
> **Q2. In the Inter-Cluster Alignment loss, the roles of $\left(a,p,n \right)$ are not sufficiently clarified. Their specific definitions and underlying physical interpretations should be explicitly described.**
>
> A2: Thank you for the helpful comment. In our framework, $\left(a,p,n \right)$ refers to anchor, positive, and negative samples whose fused embeddings ($\mathbf{l}_i$) are used for the triplet-based Inter-Cluster Alignment loss. Their definitions are as follows.
> 1. Anchor ($a$): Sample $i$ which is currently used as the basis to build $\left(a,p,n \right)$.
> 2. Positive ($p$): Another sample whose pseudo label is identical to that of the anchor, meaning the two are predicted to belong to the same cluster.
> 3. Negative ($n$): Another sample whose pseudo label differs from that of the anchor.
>
> Conceptually, these triplets characterize how clusters should be shaped in latent space. The Inter-Cluster Alignment loss (Equation (11)) imposes two complementary forces:
> 1. Attractive force: Pulling the anchor and its positives toward each other to enforce intra-cluster compactness.
> 2. Repulsive force: Pushing the anchor and its negatives away from each other to ensure inter-cluster separation.
>
> By treating every sample as an anchor during training, the Inter-Cluster Alignment triplet loss continuously refines the latent space such that samples predicted to be in the same cluster move closer together, while those predicted to belong to different clusters become more separable. This process strengthens the overall clustering structure.

---

> ### Author Response · Authors · 2025-11-19
>
> **Q3. Apart from the balancing parameter in the model, the descriptions of other hyperparameter settings are insufficient. It remains unclear whether these parameters are kept consistent across datasets, how their values are determined, and whether the comparisons with baseline methods are conducted under fair and comparable conditions.**
>
> A3: Thank you for raising this point. Except for balancing parameters in our model, other hyperparameters—such as the number of neighbors (k) and the learning rate—are not fixed across datasets. This is because different datasets have varying sample sizes, feature distributions, and information sparsity levels, which affect the clustering behavior. For each dataset, we therefore select these hyperparameters from a candidate range to obtain a better performance.
>
> Regarding the baseline methods, we strictly adhere to fair comparison principles. For each baseline, we first adopt the hyperparameter settings recommended in their original papers or in the released code. Then we implement multiple settings by performing a light search around these default values to avoid underperforming results caused by mismatched settings. Finally, we report the best performance of each baseline model in our manuscript.

---

> ### Author Response · Authors · 2025-11-19
>
> **Q4. The experimental section would benefit from a convergence analysis.**
>
> A4: Thank you for this constructive comment. We list metrics and losses over training iterations with average (avg) and standard deviation (std) from five random-seed experiments on HW and BBC to demonstrate the robustness of our model. For brevity and clarity, we report values at every ten epochs in the following table. Results show that metrics and losses stabilize with the training cycle, indicating that the model shows good convergence properties.
>
> |              |                     |                 |               | HW              |                |                 |               |                 |
> |:----:|:----:|:----:|:----:|:----:|:----:|:----:|:----:|:----:|
> |     Epoch    |     **Total   Loss**    |                 |     **ACC**       |                 |     **NMI**        |                 |    **Purity**    |                 |
> |              |     avg             |     std         |     avg       |     std         |     avg        |     std         |     avg       |     std         |
> |     1        |     42.2650        |     0.4570    |     0.9469    |     0.0210    |     0.8974     |     0.0277    |     0.9469    |     0.0210    |
> |     11       |     36.6099        |     0.0342    |     0.9493    |     0.0061    |     0.8974     |     0.0277    |     0.9493    |     0.0061    |
> |     21       |     34.9734        |     0.0795    |     0.9519    |     0.0373    |     0.8974     |     0.0277    |     0.9519    |     0.0373    |
> |     31       |     34.3469        |     0.0585    |     0.9387    |     0.0353    |     0.8961    |     0.0198    |     0.9387    |     0.0353    |
> |     41       |     33.9280        |     0.1095    |     0.9445    |     0.0649     |     0.8961    |     0.0198    |     0.9445    |     0.0649     |
> |     51       |     33.6999        |     0.0821    |     0.9722    |     0.0075     |     0.8961    |     0.0198    |     0.9722    |     0.0075     |
> |     61       |     33.5572        |     0.0753    |     0.9719    |     0.0067    |     0.9114     |     0.0045     |     0.9719    |     0.0067    |
> |     71       |     33.4697         |     0.0659     |     0.9718    |     0.0113      |     0.9114     |     0.0045     |     0.9718    |     0.0113      |
> |     81       |     33.3574        |     0.0496    |     0.9738    |     0.0073    |     0.9114     |     0.0045     |     0.9738    |     0.0073    |
> |     91       |     33.2740        |     0.0450    |     0.9741    |     0.0066    |     0.9055    |     0.0102    |     0.9741    |     0.0066    |
> |     101      |     33.2141         |     0.0405    |     0.9741    |     0.0066    |     0.9055    |     0.0102    |     0.9741    |     0.0066    |
>
> |              |                     |                 |                | BBC             |                |                 |                |                 |
> |:----:|:----:|:-----:|:----:|:----:|:----:|:----:|:----:|:----:|
> |     Epoch    |     **Total   Loss**    |                 |     **ACC**        |                 |     **NMI**        |                 |     **Purity**     |                 |
> |              |     avg             |     std         |     avg        |     std         |     avg        |     std         |     avg        |     std         |
> |     1        |     50.3218        |     0.6760    |     0.7019     |     0.0512     |     0.4038    |     0.0678    |     0.7054    |     0.0498     |
> |     11       |     40.0105        |     0.2827    |     0.8569    |     0.0391    |     0.6605    |     0.0578    |     0.7054    |     0.0498     |
> |     21       |     38.3123        |     0.1428    |     0.8593    |     0.0343    |     0.6609    |     0.0530    |     0.7054    |     0.0498     |
> |     31       |     37.2228         |     0.0995    |     0.8604    |     0.0332    |     0.6622     |     0.0516    |     0.8376    |     0.0514    |
> |     41       |     36.3464        |     0.0811    |     0.8604    |     0.0332    |     0.6622     |     0.0516    |     0.8376    |     0.0514    |
> |     51       |     35.6620        |     0.0881    |     0.8604    |     0.0332    |     0.6622     |     0.0516    |     0.8376    |     0.0514    |
> |     61       |     35.1384        |     0.0734     |     0.8604    |     0.0332    |     0.6622     |     0.0516    |     0.8540    |     0.0419    |
> |     71       |     34.7567        |     0.0594    |     0.8604    |     0.0332    |     0.6622     |     0.0516    |     0.8540    |     0.0419    |
> |     81       |     34.5532        |     0.0430    |     0.8604    |     0.0332    |     0.6622     |     0.0516    |     0.8540    |     0.0419    |
> |     91       |     34.4642        |     0.0400    |     0.8604    |     0.0332    |     0.6622     |     0.0516    |     0.8569    |     0.0391    |
> |     101      |     34.3854        |     0.0409    |     0.8604    |     0.0332    |     0.6622     |     0.0516    |     0.8569    |     0.0391    |

---

### Official Review · Reviewer_A3FV · 2025-10-30

**Soundness:** 3
**Presentation:** 2
**Contribution:** 3
**Rating:** 4
**Confidence:** 5

**Summary:**

This work addresses the challenge of effectively integrating semantic and structural information in multi-view clustering, where existing methods often emphasize one type of information while neglecting the other. To tackle this, the authors propose a dual-branch design (a VAE branch for semantic representations and a GCN branch for structure-aware features) to disentangle the two types of information and adaptively fuse them via a gated mechanism. Additionally, a triple-granularity feature alignment strategy is introduced to enforce consistency across views, samples, and clusters, enabling the model to learn clustering-friendly feature representations and improve clustering performance.

**Strengths:**

1. The proposed method introduces a novel feature disentanglement and cross-view fusion strategy, explicitly modeling the rich and complementary information in multi-view data.

2. The introduction clearly motivates the proposed approach, and the cited references are representative and sufficiently comprehensive.

3. The workflow diagram provided in the manuscript is clear and readable, aiding in the understanding of the method.

**Weaknesses:**

1. In Equation (4), the method for initializing the graph structures is not specified; it is unclear how the "Initialize graph structures" step is performed. Furthermore, the manuscript does not discuss how different graph construction methods affect the model’s performance.

2. There are inconsistencies in the coefficients: Equation (1) uses $1/N$, whereas Equation (4) uses $1/N^2$, and the mechanism for determining these coefficients is not explained. A similar inconsistency appears in the two forms of Equation (9). Additionally, $N$ is not defined.

3. The explanation of Equation (10) is unclear and difficult to understand.

4. The presentation of the Ablation Studies section lacks clear organization and logical flow. It is recommended that the authors further refine and polish this section to improve readability and coherence.

**Questions:**

In Equation (5), the overall encoding loss combines semantic and structural components. Why are their contributions considered equally important? Shouldn’t there be a hyperparameter to balance them?

---

> ### Author Response · Authors · 2025-11-19
>
> We sincerely appreciate Reviewer A3FV for the constructive comments and careful evaluation of our work. We have addressed every point in detail and hope that the revisions meet the reviewer’s expectations.
>
> **Q1. In Equation (4), the method for initializing the graph structures is not specified; it is unclear how the "Initialize graph structures" step is performed. Furthermore, the manuscript does not discuss how different graph construction methods affect the model’s performance.**
>
> A1: Thank you for this constructive comment. We leverage two graph structure initialization methods in our experiment for datasets of different kind.
>
> For image datasets, including ALOI100, HW, Yale and UCI, we obtain the k-nearest neighbors by computing Euclidean distance to construct graphs for each view feature matrix $\mathbf{X}^{(v)}$. This approach is efficient for dense image features and leverages geometric distance which is natural for visual descriptors.
>
> For text datasets, including NGS and BBC, we build a cosine-similarity-based k-nearest graph. Specifically, first, we L2-normalize each view feature $\mathbf{X}^{(v)}$; next, we compute the cosine similarity matrix and set its diagonal to zero to avoid self-loops; finally, the top-k largest similarity entries are kept as neighbors, producing a graph. This procedure is robust for sparse while high-dimensional textual representations and explicitly controls local connectivity.
>
> To further demonstrate the influence of graph construction strategies on model performance, we conducted an experiment comparing cosine-similarity graphs and Euclidean-distance graphs under the same hyperparameter settings on two representative datasets HW and BBC. The results are summarized in the table below. It can be observed that while modifying the graph construction strategy for HW leads to only a slight performance drop, applying the same change to BBC results in a substantial degradation. This indicates that text-based datasets, such as BBC, are highly sensitive to the graph construction method and particularly benefit from cosine-similarity-based graph construction method.
>
> |     Datasets    |              **ACC**            |                                  |              **NMI**            |                                  |            **Purity**           |                                  |
> |:---------------:|:-------------------------------:|:--------------------------------:|:-------------------------------:|:--------------------------------:|:-------------------------------:|:--------------------------------:|
> |                 |     cosine-similarity graphs    |     Euclidean-distance graphs    |     cosine-similarity graphs    |     Euclidean-distance graphs    |     cosine-similarity graphs    |     Euclidean-distance graphs    |
> |        HW       |               97.50             |               97.80              |               94.73             |               95.05              |               97.50             |               97.80              |
> |        BBC      |               90.07             |               24.23              |               72.75             |                1.31              |               90.07             |               33.87              |
>
> **Q2. There are inconsistencies in the coefficients: Equation (1) uses $ \frac{1}{N} $, whereas Equation (4) uses $ \frac{1}{N^2} $, and the mechanism for determining these coefficients is not explained. A similar inconsistency appears in the two forms of Equation (9). Additionally, $N$ is not defined.**
>
> A2: Thank you for raising this point. The difference in denominators is due to the difference in objects of normalization.
>
> For Equation (1) and (4), $L_\mathrm{recon}$ uses $ \frac{1}{N} $and $L_\mathrm{Structure}$ uses $ \frac{1}{N^2} $ because $X$ is a set of vectors and $A$ is a set of scalars. In calculation, $\||\hat{\mathbf{X}}^{(v)} - \mathbf{X}^{(v)} \||_2^2$ should be the sum of $N$ vectors, and $\||\hat{\mathbf{A}}^{(v)} - \mathbf{A}^{(v)} \||_2^2$ should be the sum of $N^2$ scalars. However, in fact, the implementation code is the same, both being torch.nn.functional.mse_loss ().
>
> Equation (9) has similar reasons. $L_\mathrm{distill}^{\mathrm{Semantics}}$ uses $ \frac{1}{N} $ and $L_\mathrm{distill}^{\mathrm{Structure}}$ uses $ \frac{1}{N^2} $ because $M$ is a set of vectors and $A$ is a set of scalars. In calculation, $\||\mathbf{M}^{(v)} - \mathbf{M}^* \||_2^2$ should be the sum of $N$ vectors, while $\||\hat{\mathbf{A}}^{(v)} - \mathbf{A}^* \||_2^2$ should be the sum of $N^2$ scalars. But in fact, the implementation code is the same, both being torch.nn.functional.mse_loss ().
>
> As for $N$, it is the total number of samples in the experimented dataset.

---

> ### Author Response · Authors · 2025-11-19
>
> **Q3. The explanation of Equation (10) is unclear and difficult to understand.**
>
> A3: Thank you for pointing this out. The Equation (10) is designed to ensure consistency between the fused embedding $\mathbf{l}_i$ and its semantic and structure-aware counterparts $\boldsymbol{\mu}_i^{(v)}$ and $\mathbf{h}_i^{(v)}$.
>
> Specifically, for each sample $i$, we treat $\left(\mathbf{l}_i, \boldsymbol{\mu}_i^{(v)} \right)$ and $\left(\mathbf{l}_i, \mathbf{h}_i^{(v)} \right)$ as positive pairs, which form the numerator of Equation (10). Semantic and structure-aware embeddings from all other samples serve as negative information for sample $i$, forming the denominator. In optimization, we aim to minimize Equation (10), i.e., increasing the similarity between $\left(\mathbf{l}_i, \boldsymbol{\mu}_i^{(v)}\right)$ and $\left(\mathbf{l}_i, \mathbf{h}_i^{(v)} \right)$  (the numerator), and simultaneously reducing the similarity between $\mathbf{l}_i $ and embeddings from other samples.
>
> This design encourages the fused representation $\mathbf{l}_i $ to preserve complementary semantic–structural information from all views, and maintain global discriminability across samples, which is essential for clustering.
>
> **Q4. The presentation of the Ablation Studies section lacks clear organization and logical flow. It is recommended that the authors further refine and polish this section to improve readability and coherence.**
>
> A4: Thank you for this meaningful comment. Following the comment, we reorganized and refined the Ablation Studies section to improve clarity and logical coherence. The revised version clearly separates the analysis into four modules (the Semantics Encoding Module, Structure Encoding Module, Gated Feature Fusion Module, and Feature Alignment Module), and presents the results in a more structured and interpretable manner. The rewritten section is provided below:
>
> To clearly illustrate the contribution of each core component in DREAM, we conduct a systematic ablation study by removing the Semantics Encoding Module, Structure Encoding Module, Gated Feature Fusion Module, and Feature Alignment Module individually. Table 3 reports the performance under each ablation setting.
>
> We first examine the two encoding branches. Removing the Semantics Encoding Module leads to a consistent drop in performance across datasets, indicating that semantic representations provide indispensable, discriminative cues for clustering. Removing the Structure Encoding Module results in the most severe performance degradation—for example, ACC decreases by 20% on the UCI dataset—demonstrating that inter-sample structural relations are fundamental for reliable multi-view clustering.
>
> Next, we assess the importance of the fusion mechanism by replacing the Gated Feature Fusion Module with simple averaging. While the changed model occasionally surpasses single-branch variants, it remains substantially inferior to the full DREAM model. This indicates that naive averaging fails to effectively leverage the complementary information of semantic and structural embeddings, whereas gated fusion adaptively balances view informativeness.
>
> Finally, disabling the Feature Alignment Module also results in noticeable performance degradation. On the UCI dataset, ACC decreases from 95.90% to 88.35%, demonstrating that triple-granularity alignment strengthens cluster cohesion by harmonizing representations across views, samples, and clusters.
>
> Overall, the ablation results validate that each module contributes meaningfully and that the integration of dual-branch encoding, gated feature fusion, and feature alignment achieves superior multi-view clustering performance.

---

> ### Author Response · Authors · 2025-11-19
>
> **Q5. In Equation (5), the overall encoding loss combines semantic and structural components. Why are their contributions considered equally important? Shouldn’t there be a hyperparameter to balance them?**
>
> A5: Thank you for the insightful comment. Our decision not to introduce an additional balancing hyperparameter in Equation (5) can be explained as follows:
> 1. Semantic and structural losses naturally operate on comparable scales. The semantic and structural branches are designed to capture complementary aspects of multi-view data—instance-specific semantics and inter-sample relational structure. During training, we observe that the magnitudes of the two losses remain within a similar range (typically within a 1-4× ratio), with neither term dominating or collapsing. Therefore, equal weighting preserves the intended cooperative contribution of the two types of cues.
> 2. Avoiding unnecessary hyperparameter inflation. Introducing an extra balancing hyperparameter in Equation (5) did not bring performance gains in our preliminary trials. Since the framework already includes several hyperparameters for balancing loss terms (e.g., Equation (16)), adding an additional coefficient would increase tuning complexity without yielding observable benefits. We thus adopt the simpler and more stable formulation.
> 3. Additional empirical verification. To further validate this design choice, we introduce a balancing hyperparameter for the structural term $L_\mathrm{Structure}$ in Equation (5), i.e.,
>  $L_\mathrm{Encode} = L_\mathrm{Semantics} + \varpi L_\mathrm{Structure}$, and conduct experiments on the HW and BBC datasets. The results, reported in the table below, show that adjusting $\varpi$ yields no performance improvement, confirming that introducing a hyperparameter to balance the two losses in Equation (5) is unnecessary.
>
> |      | $\varpi=0.001$ | $\varpi=0.01$ | $\varpi=0.1$ |  $\varpi=1$  |  $\varpi=10$ | $\varpi=100$ | $\varpi=1000$ |
> |:--------:|:-------:|:------:|:-----:|:-----:|:-----:|:-----:|:------:|
> |       |         |        |       |   **ACC**    |       |       |        |
> |    HW    |  97.60  |  97.75 | 97.55 | 97.80 | 97.65 | 96.25 |  96.25 |
> |    BBC   |  88.32  |  88.18 | 88.32 | 90.07 | 88.61 | 78.39 |  75.62 |
> |      |         |        |       |   **NMI**    |       |       |        |
> |    HW    |  94.69  |  94.96 | 94.58 | 95.05 | 94.90 | 91.83 |  91.83 |
> |    BBC   |  68.77  |  69.23 | 68.67 | 72.75 | 69.81 | 55.72 |  47.42 |
> |  |         |        |       |   **Purity**    |       |       |        |
> |    HW    |  97.60  |  97.75 | 97.55 | 97.80 | 97.65 | 96.25 |  96.25 |
> |    BBC   |  88.32  |  88.18 | 88.32 | 90.07 | 88.61 | 78.39 |  75.62 |

---

### Official Review · Reviewer_hicV · 2025-10-30

**Soundness:** 3
**Presentation:** 3
**Contribution:** 4
**Rating:** 6
**Confidence:** 5

**Summary:**

The paper develops a multi-view clustering framework based on a dual-branch representation that simultaneously captures semantic and structural information from multiple views. It employs a gated fusion module that adaptively balances the contributions of semantic and structural features according to the characteristics of the data, producing latent representations more suitable for clustering. In addition, the authors introduce a triple-granularity feature alignment mechanism to enforce consistency at three levels. This design enhances the coherence between semantic and structural information while improving inter-cluster separability.

**Strengths:**

1.The developed framework systematically tackles imbalanced integration of semantic and structural information, conflicts in feature fusion, and limited feature alignment, providing a well-motivated solution to longstanding issues in deep MVC.
2.The paper’s division of multi-view clustering methods into semantics- and structure-oriented categories is insightful, highlighting the need to jointly leverage both information types.

**Weaknesses:**

1.The rationale for using structural cues in the gated fusion module as complementary guidance for cross-view embedding fusion is unclear. Clarification with semantic-guided fusion would be beneficial.
2.Some acronyms, such as VAE and GCN, are introduced without explanation. Providing their full names on first mention would improve clarity.
3.Certain modules in Figure 1 are not fully described, making it difficult to understand their exact functionality.
4.Regarding the graph reconstruction loss, which minimizes the mean squared error between predicted and ground-truth adjacency matrices, it is not explained whether using the ground-truth adjacency directly would suffice. If so, the role of the GCN encoder requires further justification.
5.Both Equation (3) and Equation (14) involve KL divergence, but the distinction between them is not clearly explained.
6.In the experiments, it would be helpful to include references or links for the datasets used to enhance reproducibility.

**Questions:**

Please see the Weaknesses section.

---

> ### Author Response · Authors · 2025-11-19
>
> We greatly appreciate the constructive comments provided by Reviewer hicV for the revision of this manuscript. We have carefully addressed all concerns in detail and hope that all issues have been satisfactorily resolved.
>
> **Q1. The rationale for using structural cues in the gated fusion module as complementary guidance for cross-view embedding fusion is unclear. Clarification with semantic-guided fusion would be beneficial.**
>
> A1: Thank you for this constructive comment. The rationale in using structural cues in the gated fusion module as complementary guidance for cross-view embedding fusion can be demonstrated in the following:
> 1. The structural embeddings $\mathbf{h}_i^{(v)}$ incorporates inter-instance information, thus hints the structural reliability of sample $i$ in view $v$. In other words, it captures how consistently this sample aligns with its local neighborhood in that view’s graph structure. This property provides valuable cues for view weighting during fusion.
> 2. We conducted an experiment replacing the structural cues with semantic cues in the gated-fusion model to verify the effectiveness of structural cue-aided gated fusion. Results are listed in the table below, showing that clustering performance is better when structural cues are employed as the complementary guidance for the gated-fusion model.
>
> |     Datasets    |     Ours (with structural-guided fusion)    |     Ours (with semantic-guided fusion)    |
> |:-----------------:|:---------------------------------------------:|:-------------------------------------------:|
> |                 | **ACC**                                     |                                           |
> |     Yale        |     78.18                                   |     76.97                                 |
> |     NGS         |     97.80                                   |     86.00                                 |
> |     BBC         |     90.07                                   |     87.74                                 |
> |     UCI         |     95.90                                   |     93.85                                 |
> |     HW          |     97.80                                   |     97.55                                 |
> |     ALOI100     |     87.00                                   |     85.99                                 |
> |                 | **NMI**                                     |                                           |
> |     Yale        |     79.87                                   |     76.60                                 |
> |     NGS         |     92.90                                   |     73.04                                 |
> |     BBC         |     72.75                                   |     68.25                                 |
> |     UCI         |     92.01                                   |     89.75                                 |
> |     HW          |     95.05                                   |     94.39                                 |
> |     ALOI100     |     90.88                                   |     90.86                                 |
> |                 | **Purity**                                  |                                           |
> |     Yale        |     82.42                                   |     80.00                                 |
> |     NGS         |     98.00                                   |     88.40                                 |
> |     BBC         |     90.07                                   |     87.74                                 |
> |     UCI         |     96.30                                   |     95.10                                 |
> |     HW          |     97.80                                   |     97.55                                 |
> |     ALOI100     |     88.18                                   |     89.56                                 |
>
> **Q2. Some acronyms, such as VAE and GCN, are introduced without explanation. Providing their full names on first mention would improve clarity.**
>
> A2: Thank you for pointing this out. The full name of VAE is “Variational Autoencoder”, and the full name of GCN is “Graph Convolutional Network”. We will provide full names of VAE and GCN on the first mention in our revised version.

---

> ### Author Response · Authors · 2025-11-19
>
> **Q3. Certain modules in Figure 1 are not fully described, making it difficult to understand their exact functionality.**
>
> A3: Thank you for this valuable comment. The functionality of each module can be described below in order:
>
> First, the multi-view dataset $\mathcal{X}$ is processed by the Semantics Encoding Branch (the upper left module with grey background in Figure 1) and Structure Encoding Branch (the middle left module with grey background) separately, obtaining latent feature $\boldsymbol{\mu}_i^{(v)}$ and $\mathbf{h}_i^{(v)}$ for each sample $i$.
>
> Second, $\boldsymbol{\mu}_i^{(v)}$ and $\mathbf{h}_i^{(v)}$ are processed by the Gated Feature Fusion Module (the module with light orange background) to obtain the fused latent feature $\mathbf{l}_i$ . Specifically, for each view $v$, $\boldsymbol{\mu}_i^{(v)}$ and $\mathbf{h}_i^{(v)}$ are first fused using a learned gating strategy to produce a latent embedding $\mathbf{g}_i^{(v)}$, which integrates information from both semantic and structural representations. Then, the embeddings $\mathbf{g}_i^{(v)}$ across all views are combined using cross-view weighting cues derived from $\mathbf{h}_i^{(v)}$, resulting in the final fused feature $\mathbf{l}_i$.
>
> Third, $\mathbf{l}_i$ is processed by the clustering module (the bottom right part in the module with orange background), obtaining clustering result.
>
> Fourth, the module with light blue background comprises three alignment strategies corresponding to the first, second, and third step. The Cross-View Alignment module encourages each view of sample $i$ to capture more consistent semantic and structural information. The Intra-Sample Alignment module encourages the fused embedding $\mathbf{l}_i$ to remain close to its semantic and structure-aware counterparts ($\boldsymbol{\mu}_i^{(v)}$ and $\mathbf{h}_i^{(v)}$). Finally, the Inter-Cluster Alignment module enhances the discriminability among different clusters.

---

> ### Author Response · Authors · 2025-11-19
>
> **Q4. Regarding the graph reconstruction loss, which minimizes the mean squared error between predicted and ground-truth adjacency matrices, it is not explained whether using the ground-truth adjacency directly would suffice. If so, the role of the GCN encoder requires further justification.**
>
> A4: Thank you for this insightful comment. To verify the necessity of the GCN encoder, we conducted an experiment in which the GCN encoder is replaced by a linear layer that directly receives the ground-truth adjacency matrix as input. The linear layer is employed because the next module requires inputs of the same size. The results, summarized in the table below, show that while using the ground-truth adjacency achieves reasonable (good though not better) performance on image datasets (Yale, UCI, HW, and ALOI100), it leads to a significant performance drop on text-based datasets (NGS and BBC). These findings demonstrate that the GCN encoder is essential for effectively dealing with both image- and text-based datasets simultaneously.
>
> |     Datasets    |     Ours with GCN encoder    |     Ours with ground-truth   adjacency replacing GCN encoder    |
> |:-----------------:|:------------------------------:|:-----------------------------------------------------------------:|
> |                 | **ACC**                      |                                                                 |
> |     Yale        |     78.18                    |     77.58                                                       |
> |     NGS         |     97.80                    |     56.60                                                       |
> |     BBC         |     90.07                    |     51.24                                                       |
> |     UCI         |     95.90                    |     92.90                                                       |
> |     HW          |     97.80                    |     94.50                                                       |
> |     ALOI100     |     87.00                    |     81.19                                                       |
> |                 | **NMI**                      |                                                                 |
> |     Yale        |     79.87                    |     78.28                                                       |
> |     NGS         |     92.90                    |     31.93                                                       |
> |     BBC         |     72.75                    |     27.00                                                       |
> |     UCI         |     92.01                    |     89.56                                                       |
> |     HW          |     95.05                    |     92.31                                                       |
> |     ALOI100     |     90.88                    |     89.03                                                       |
> |                 | **Purity**                   |                                                                 |
> |     Yale        |     82.42                    |     80.61                                                       |
> |     NGS         |     98.00                    |     61.20                                                       |
> |     BBC         |     90.07                    |     59.56                                                       |
> |     UCI         |     96.30                    |     95.25                                                       |
> |     HW          |     97.80                    |     97.15                                                       |
> |     ALOI100     |     88.18                    |     84.85                                                       |
>
> **Q5. Both Equation (3) and Equation (14) involve KL divergence, but the distinction between them is not clearly explained.**
>
> A5: Thank you for raising this point. Equation (3) is leveraged in the Semantics Encoding Branch to encourage the latent variable distribution of the encoder output to approach the standard normal distribution, making the latent variable space more continuous and smoother, while Equation (14) is leveraged in the Clustering Module to make the soft assignment of the sample closer to the target clustering with high confidence.
>
> **Q6. In the experiments, it would be helpful to include references or links for the datasets used to enhance reproducibility.**
>
> A6: Thank you for this helpful comment. We will add links for datasets used in the experiment in the final version of our paper.

---

### Comment · Area_Chair_NBMK · 2025-12-01

Dear Authors,

Reviewer A3FV (who gives the only negative score) raised concerns regarding the graph-structure initialization in Eq.4 and its impact on performance, as well as the inconsistency of coefficients between Eq.1 and Eq.4, with the behind mechanism not explained. Please clarify these issues again to help me better evaluate this submission.

I also check this manuscript and notice that deep NN-based methods often struggle to achieve reliable clustering results for small-scale datasets and that with a large number of classes, e.g., Yale, BBC, ALOI100. However, the proposed method has achieved a significant clustering improvement for these datasets compared to previous deep clustering methods, and the reasons for this need to be further explained and highlighted. Additionally, the difference between this work and some related ones [Ref1, Ref2] needs to be discussed:

[Ref1] COPER: Correlation-based Permutations for Multi-View Clustering, ICLR 2025

[Ref2] Performance Gaps in Multi-view Clustering under the Nested Matrix-Tensor Model, ICLR 2024

Best regards,

AC

---

> ### Author Response · Authors · 2025-12-03
>
> Dear AC,
>
> We sincerely appreciate your dedicated efforts throughout the review process, and we are also grateful to all four Reviewers for their recognition of our work and their constructive feedback. We have carefully addressed concerns you mentioned, and hope that our clarification can resolve them satisfactorily.
>
>
> **Q1: Further clarification regarding the concerns of reviewer A3FV (who gives the only negative score) about the graph structure initialization in Eq. 4 and its impact on performance.**
>
> A1: Thank you for raising this point. We clarify that our method adopts data-type-appropriate initialization methods to extract meaningful structural representations for different datasets.
>
> 1. We use two initialization strategies for image datasets and text datasets, separately. Image-view features (ALOI100, HW, Yale, and UCI) exhibit dense and continuous distributions, and Euclidean distance reliably reflects local geometric similarity. Thus, we use Euclidean distance to initialize k-nearest neighbors graph (k-NNG) for image datasets. Textual features (BBC, NGS), however, are high-dimensional and sparse, and cosine similarity is well known to better capture semantic proximity. Thus, we initialize k-NNG using cosine similarity for text datasets.
> 2. To examine graph-structure initialization strategy's impact on model performance, we conducted a controlled comparison of the two graph-construction methods under identical settings on a representative image dataset (HW) and a representative text dataset (BBC). From the results, it is clear that for HW (image dataset), replacing Euclidean distance with cosine similarity causes only a minor performance change, indicating that our approach remains robust regardless of the choice of graph-structure initialization strategy for image datasets, while for BBC (text dataset), replacing cosine similarity with Euclidean distance leads to a substantial performance drop, confirming that cosine similarity is essential for constructing reliable graphs for textual datasets. These results demonstrate that using an appropriate similarity metric for different data modality is crucial, and inappropriate similarity metric would affect model performance.
>
> |     Datasets    |              **ACC**            |                                  |              **NMI**            |                                  |            **Purity**           |                                  |
> |:---------------:|:-------------------------------:|:--------------------------------:|:-------------------------------:|:--------------------------------:|:-------------------------------:|:--------------------------------:|
> |                 |     cosine-similarity graphs    |     Euclidean-distance graphs    |     cosine-similarity graphs    |     Euclidean-distance graphs    |     cosine-similarity graphs    |     Euclidean-distance graphs    |
> |        HW       |               97.50             |               97.80              |               94.73             |               95.05              |               97.50             |               97.80              |
> |        BBC      |               90.07             |               24.23              |               72.75             |                1.31              |               90.07             |               33.87              |

---

> ### Author Response · Authors · 2025-12-03
>
> **Q2: A clarification regarding the mechanism behind the coefficient discrepancy between Eq. 1 and Eq. 4.**
>
> A2: Thank you for pointing out this issue. We use different coefficients in the two equations  because each loss term normalizes over a different number of elements, depending on the structure of the matrix being processed.
>
> 1. Why Eq.1 uses $\frac{1}{N}$ while Eq.4 uses $ \frac{1}{N^2} $. The equations first compute a squared reconstruction error between each pair of elements with the same subscript; next, they sum the errors over all elements and normalize by the number of elements to obtain the average error for that view; finally, the averaged errors from all views are summed to obtain the overall loss. Using $\frac{1}{N}$ or $\frac{1}{N^2}$ depends on the number of elements being computed.
> For Eq.1, $\mathbf{X}^{(v)}$ (where $\mathbf{X}^{(v)} = $ { $\mathbf{x}\_1^{(v)}, \mathbf{x}\_2^{(v)}, \dots, \mathbf{x}\_N^{(v)} $ } $\in \mathbb{R}^{N \times d^{(v)}}$)  contains $N$ elements (sample vectors). The squared error sums over $N$ vector differences. Therefore, we use $N$ as the mean normalization denominator.
> For Eq.4, $\mathbf{A}^{(v)}$ (where $\mathbf{A}^{(v)} = $ { $a^{(v)}_{ij} $ } $\in \mathbb{R}^{N \times N}$) consists of $N \times N$ elements (scalars). The squared error sums over $N \times N$ scalar differences, hence ${N^2}$ is used as the mean normalization denominator.
> To summarize, the denominators reflect the number of elements involved in each loss term.
> 1. In practice, the two losses are implemented using torch.nn.functional.mse_loss(), which automatically computes the mean over all elements. Thus, the normalization behavior of the equations is consistent with the actual implementation.
>
> **Q3: I also check this manuscript and notice that deep NN-based methods often struggle to achieve reliable clustering results for small-scale datasets and that with a large number of classes, e.g., Yale, BBC, ALOI100. However, the proposed method has achieved a significant clustering improvement for these datasets compared to previous deep clustering methods, and the reasons for this need to be further explained and highlighted.**
>
> A3: Thank you for the insightful comment. We believe that the difficulty faced by deep neural network-based clustering methods on small-scale datasets (e.g., Yale, BBC) and datasets with a large number of fine-grained classes (e.g., ALOI100) primarily stems from representation unreliability and insufficient class-specific information. When only a small number of samples are available per class, it is difficult for the model to extract enough discriminative cues, and the learned embeddings may contain misleading signals or lack informative structure, ultimately leading to suboptimal clustering performance. Our method achieves improvements on these challenging datasets for the following reasons:
>
> 1. Dual-branch disentanglement with adaptive feature fusion enhances information richness and representation reliability. Unlike methods that extract a single type of information and embed it into one latent space, our framework explicitly disentangles semantic information (via the VAE branch) and structural information (via the GCN branch), and integrates them through a gated fusion mechanism. This design not only compels the model to learn complementary perspectives of the data, thereby enriching information diversity, but also enables it to dynamically balance the semantic and structural contributions according to the informativeness of each, thereby improving the reliability of the learned representations.
>
> 2. Triplet-alignment improves robustness against fine-grained noise. Datasets such as ALOI100 contain fine-grained categories, leading to subtle inter-class differences. Consequently, representations are more susceptible to noise. Our alignment mechanism jointly aligns latent spaces across views, across information types, and within clusters, forcing the model to aggregate information from multiple sources, maintaining consistent representations while mitigating noise.

---

> ### Author Response · Authors · 2025-12-03
>
> **Q4: Additionally, the difference between this work and some related ones [Ref1, Ref2] needs to be discussed:**
> **[Ref1] COPER: Correlation-based Permutations for Multi-View Clustering, ICLR 2025**;
> **[Ref2] Performance Gaps in Multi-view Clustering under the Nested Matrix-Tensor Model, ICLR 2024**
>
> A4: Thank you for guiding us to these related works. We summarize the key differences between our work and the two related papers below.
>
> "COPER: Correlation-based Permutations for Multi-View Clustering" (ICLR 2025) proposes an end-to-end deep multi-view clustering framework that builds fused representations via a novel permutation-based canonical-correlation objective, and then learns cluster assignments by identifying consistent pseudo-labels across views. The paper further provides theoretical analysis on how the learned embeddings approximate those obtained by supervised LDA and establishes a theoretical bound on pseudo-label-caused error.  While COPER focuses on semantic features, learning fused embeddings across views via a CCA-style objective enhanced by permutation and presenting pseudo-labeling procedure for identifying consistent labels across views, our work targets a different set of problems. We categorize information in multi-view datasets into semantic and structure-aware groups and explicitly disentangle them through dedicated encoders (VAE for semantics, GCN for structure) with distillation applied to maintain key information within each information source. To integrate two source of information, we further introduce a reliability-aware gated fusion and  triple-granularity alignment scheme.  To summarize, COPER emphasizes correlation-based semantic feature fusion and pseudo-label consistency across views to build an end-to-end multi-view clustering framework for general data types, whereas our method emphasizes explicit semantic-structural disentanglement, adaptive fusion and triple-granularity alignment to improve information richness and handle modality-dependent reliability.
>
>
> "Performance Gaps in Multi-view Clustering under the Nested Matrix-Tensor Model" (ICLR 2024) focuses on a statistical-computational target, quantifying the performance gap between a tensor approach and an unfolding (matrix) approach under a nested matrix-tensor model. Our work differs fundamentally in scope and goals. By developing a practical deep-learning framework for multi-view clustering, we contribute in algorithmic design and empirical study.
>
> In short, COPER and our method propose different multi-view clustering pipelines, and while the nested matrix-tensor analysis provides theoretical insights, our work focuses on a practical algorithmic framework.
>
> We highly value the review and revision process. We hope our responses sufficiently clarify the raised concerns, and we remain available to provide any further information if required.
>
> Best Regards,
>
> Authors

---

### Meta-Review · Area_Chair_NBMK · 2026-01-06

**Summary:**

This paper presents a multiview clustering framework that explicitly disentangles and adaptively fuses semantic and structural information via a dual-branch architecture, coupled with a triple-granularity alignment strategy. The core contributions are semantic-structural disentanglement, gated cross-view fusion, and multi-level consistency alignment, which are well-motivated and address the limitations in existing deep multiview clustering methods.

All four reviewers acknowledged the paper's soundness and technical contribution. While initial reviews raised several clarifications and concerns, particularly regarding design rationale and implementation details, the authors provided comprehensive responses that addressed key methodological questions and strengthened the empirical validation. In conclusion, this work meets the acceptance bar for ICLR in terms of novelty, technical rigor, and experimental validation. The authors also need to polish the final version by taking into account the comments of the reviewers and AC.

**Reviewer Concerns:**

Reviewer hicV: Concerns about the structural-guided fusion rationale and GCN encoder necessity were substantively addressed with new comparative experiments, elevating confidence in the method's design.

Reviewer yNg5: Questions regarding pseudo-label generation, triplet definitions, and convergence were answered in the rebuttal.

Reviewer A3FV: Core methodological questions (graph initialization, coefficient inconsistencies) were not directly addressed. These issues were further addressed during the discussion between AC and the authors.

Reviewer K17d: Clarifications on hyper-parameter, alignment loss weighting, and cross-view fusion implementation were provided in response.

**Reviewer Scores:**

The reviewer's questions were covered by the author's further qualitative analysis and quantitative results. Therefore, further discussion is likely to contribute to improving the score.

---

### Decision · Program_Chairs · 2026-01-26

Accept (Poster)